# Plasma membrane H$^+$-ATPases sustain pollen tube growth and fertilization

Robert D. Hoffmann[1,5], Maria Teresa Portes[2,5], Lene Irene Olsen[1,5], Daniel Santa Cruz Damineli[2,3,5], Maki Hayashi[1,5], Custódio O. Nunes[2], Jesper T. Pedersen[1], Pedro T. Lima[4], Cláudia Campos[4], José A. Feijó[2,4 ✉] & Michael Palmgren[1 ✉]

Pollen tubes are highly polarized tip-growing cells that depend on cytosolic pH gradients for signaling and growth. Autoinhibited plasma membrane proton (H$^+$) ATPases (AHAs) have been proposed to energize pollen tube growth and underlie cell polarity, however, mechanistic evidence for this is lacking. Here we report that the combined loss of *AHA6, AHA8,* and *AHA9* in *Arabidopsis thaliana* delays pollen germination and causes pollen tube growth defects, leading to drastically reduced fertility. Pollen tubes of *aha* mutants had reduced extracellular proton (H$^+$) and anion fluxes, reduced cytosolic pH, reduced tip-to-shank proton gradients, and defects in actin organization. Furthermore, mutant pollen tubes had less negative membrane potentials, substantiating a mechanistic role for AHAs in pollen tube growth through plasma membrane hyperpolarization. Our findings define AHAs as energy transducers that sustain the ionic circuit defining the spatial and temporal profiles of cytosolic pH, thereby controlling downstream pH-dependent mechanisms essential for pollen tube elongation, and thus plant fertility.

[1] Department for Plant and Environmental Sciences, University of Copenhagen, 1871 Frederiksberg C, Denmark. [2] Department of Cell Biology and Molecular Genetics, University of Maryland, College Park, MD 20742, USA. [3] Department of Pediatrics, Faculdade de Medicina da Universidade de São Paulo, São Paulo, SP 01246-903, Brazil. [4] Instituto Gulbenkian de Ciência, Oeiras 2780-156, Portugal. [5] These authors contributed equally: Robert D. Hoffmann, Maria Teresa Portes, Lene Irene Olsen, Daniel Santa Cruz Damineli, Maki Hayashi. ✉email: jfeijo@umd.edu; palmgren@plen.ku.dk

Flowering plants evolved a complex process to deliver sperm cells for double fertilization and seed formation. When pollen, the male gametophyte, lands on a receptive stigma of a flower, it germinates to produce a cell extension, the pollen tube, which carries non-motile sperm cells. Pollen tubes grow through the female sporophytic tissues and discharge the sperm cells into the female gametophyte, the embryo sac[1,2].

Pollen tubes elongate by tip growth, a mechanism of cell elongation in which secretory vesicles fuse with the plasma membrane exclusively at the growing apex, that is common to root hairs, filamentous fungi, and developing neurites[3,4]. Pollen tubes evolved to be streamlined chemotropic growing cells. They possess dramatically polarized cytosolic ion gradients at the tip ($H^+$, $Ca^{2+}$, $Cl^-$), which are believed to modulate the cellular processes and structures involved in tip growth, including the actin cytoskeleton[5–7]. Although cytosolic buffering mechanisms were thought to preclude the existence of cytosolic pH gradients[3,8], pollen tubes were shown to have apical gradients with up to one pH unit ranging over 10–30 μm from the extreme tip of the pollen tube[7,9,10,11]. These gradients correlate spatially with a loop of extracellular $H^+$ fluxes, in which the pollen tube tip sinks the net effluxes generated all over the tube shank, thus creating an apparent $H^+$ short-circuit[9].

The conspicuous $H^+$ effluxes along the pollen tube shank have been attributed to plasma membrane $H^+$-ATPases (AHAs for Autoinhibited $H^+$-ATPases; ref. [11]), since they constitute the main family of $H^+$ export proteins in plants[12–14]. Molecular cell biology and pharmacology analyses support the hypothesis that AHAs indeed generate extracellular $H^+$ fluxes and sustain the cytosolic $H^+$/pH gradient in growing pollen tubes[8,15–19]. For instance, in tobacco (*Nicotiana tabacum*), the isoform NtAHA1 localizes to the pollen tube plasma membrane only in the shank (the entire length of the pollen tube behind the tip-most region), being actively excluded from the tip[17], in close spatial association with the patterns of cytosolic pH, extracellular $H^+$ flux, and actin remodeling (Fig. 1 in ref. [5]). However, direct evidence linking AHAs with cellular functions was lacking due to its high degree of genetic redundancy and difficulties in manipulating pollen tube dynamics in non-model species.

Here, we show that the pollen-specific plasma membrane $H^+$-ATPase isoforms AHA6, AHA8, and AHA9 from *Arabidopsis thaliana* are essential for pollen tube growth and fertility. Triple mutants of these pumps have strongly reduced fertility due to premature pollen tube arrest, and mutant pollen tubes showed reduced extracellular ion fluxes, reduced plasma membrane potential, disturbed spatiotemporal patterns of cytosolic pH and disrupted actin organization. We conclude that the AHAs energize a loop of $H^+$ current around the tip, which is essential for pollen tube function.

## Results and discussion
### AHA6, AHA8, and AHA9 are plasma membrane proton ATPases.
The *A. thaliana* genome encodes 11 presumably functional $H^+$-ATPases (*AHA1–11*)[18], of which the three isoforms *AHA6*, *AHA8*, and *AHA9*, the closest homologs of *NtAHA1*, are predominantly expressed in pollen and pollen tubes[14,20,21] (Supplementary Fig. 1). Since genetic evidence for the involvement of plasma membrane $H^+$-ATPases in pollen tube tip growth is lacking, we investigated the role of the three pumps encoded by these genes in pollen tube growth.

To confirm the expression of *AHA6*, *AHA8*, and *AHA9* in the male gametophyte, we performed a β-glucuronidase (GUS) reporter-aided analysis of the *AHA6*, *AHA8*, and *AHA9* promoter activities. Histochemical staining of GUS activity in *AHA6* promoter *pAHA6::GUS*, *pAHA8::GUS*, and *pAHA9::GUS*

transgenic lines revealed that all three promoters were active in pollen and pollen tubes (Fig. 1a, left). When the same promoters were used to drive expression of chimeras between AHA6, AHA8, and AHA9, and green fluorescent protein (GFP), fluorescence was evident at the pollen tube shank (Fig. 1a, right), whereas the three proteins were excluded from the apex of the pollen tubes (Fig. 1a, Supplementary Fig. 2b), closely resembling the distribution of NtAHA1 in tobacco[16]. AHA6 localized along the entire length of the pollen tube, whereas AHA8 was absent from shank regions further away from the tip, and AHA9 was present only at the shank close to the pollen germination pore (Fig. 1a, Supplementary Fig. 2b).

In the *aha6/8* double mutant, *AHA6::GFP* fully complemented the loss of *AHA6* (Supplementary Fig. 4a). *AHA8::GFP* partially rescued the aberrant phenotype (whereas *AHA8* without GFP fully rescued it). *AHA9::GFP* fully rescued the *aha6/9* phenotype of reduced seed setting (Supplementary Fig. 4a).

We confirmed the $H^+$ pump activity of each of these proteins by heterologously expressing cDNA containing their coding region in the yeast strain RS-72, in which the native proton pump *PMA1* is under the control of a galactose-inducible promoter[22]. This strain can only grow in the absence of galactose when a functional plasma membrane $H^+$ pump is expressed. As a C-terminal autoinhibitory domain has been identified in plasma membrane $H^+$-ATPases[13,14], we also expressed gene versions that encode truncated pumps lacking 92 C-terminal amino acid residues. The autoinhibitory C-terminal domain of AHAs does not block the activity of the pump, but controls the degree of coupling between ATP hydrolysis and $H^+$ pumping, i.e., the pumping efficiency[23]. Thus, the ability of full-length AHAs to complement *S. cerevisiae pma1* depends on the size of the electrochemical $H^+$ gradient the pumps are pumping against and, if growth is only reduced, the period of time of the growth assay. The truncated versions of all three pumps complemented *pma1* even at an external pH of 3.5, confirming a $H^+$-extrusion pump function (Fig. 1b). Expression of full-length cDNA did not complement growth, suggesting that the autoinhibitory domain strongly reduces proton pumping in the yeast system (Fig. 1b).

We then assayed the specific ATPase activity of the C-terminally truncated pumps as a function of pH. All three gametophyte-specific pumps had their maximum activities close to pH 6.5 (Fig. 1c). This is similar to the pH optimum of AHA2, an isoform strongly expressed in the vegetative parts of the plant (the sporophyte)[24]. Moreover, all pumps had overlapping $K_m$ values for ATP (Fig. 1d), suggesting that sporophyte and gametophyte $H^+$-ATPases have similar pump properties.

### *aha6/8/9* pollen tubes have severe growth phenotypes.
We investigated the roles of *AHA6*, *AHA8*, and *AHA9* genetically by characterizing T-DNA insertion lines in *A. thaliana* (Fig. 2a). The *aha6-1*, *aha8-1*, *aha8-3*, *aha9-4*, and *aha9-5* mutant plants were confirmed to have undetectable levels of the corresponding RNA transcript in mature flowers of homozygous T-DNA insertion lines (Fig. 2b, c). Of the single mutant plants, only *aha6* showed mild phenotypes (Supplementary Table 1; Supplementary Fig. 3a–c).

Given that genetic redundancy is a well-established phenomenon in ion transporter families of pollen[25,26], we crossed the individual lines to generate homozygous double and triple mutant plants. When self-fertilized, the three combinations of double mutants (*aha6/8*, *aha6/9*, and *aha8/9*) showed different degrees of early pollen tube growth arrest (Fig. 2d, e), reduced pollen tube elongation rates (Fig. 2e), and reduced fertility (Fig. 2f). *aha6/8* pollen tubes grew at a similar rate as those of the wild-type but terminated growth much earlier, whereas *aha6/9* pollen tubes

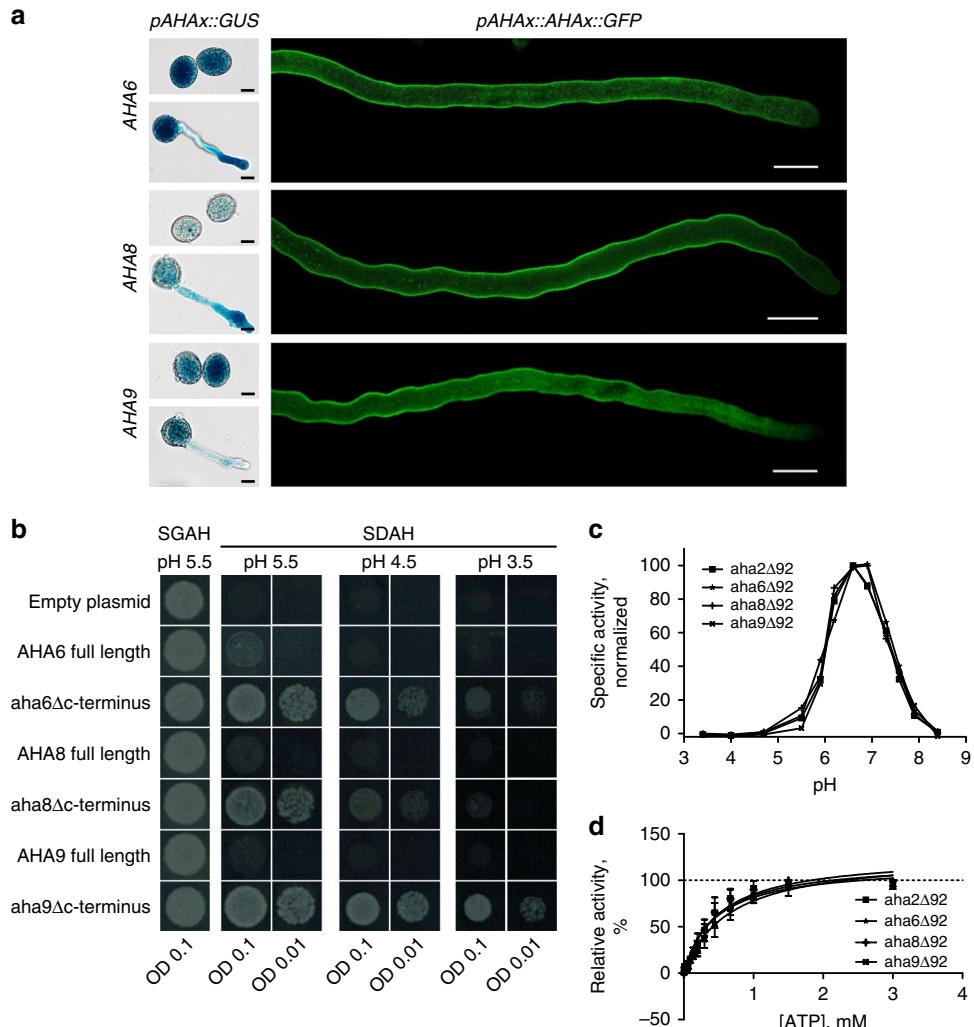

**Fig. 1 AHA6, AHA8, and AHA9 are proton pumps localized in the plasma membrane of pollen tubes. a** Left, Histochemical staining for GUS activity in pollen grains and grown pollen tubes stably expressing *AHA* promoter::GUS fusions and observed using a light microscope. Right, Pollen tubes stably expressing GFP-tagged AHA proteins under control of their endogenous promoters and observed using a confocal microscope (see also Supplementary Fig. 2b). *AHA6::GFP* and *AHA8::GFP* were expressed in the *aha6/8* mutant and AHA9::GFP was expressed in the *aha9* mutant. Scale bars = 10 μm. **b** Mutated AHA6, AHA8, or AHA9 proteins lacking the autoinhibitory C-terminal domain, but not the full-length proteins, support yeast growth in the absence of its own plasma membrane proton ATPase, PMA1. The yeast strain RS-72 used in this analysis has *PMA1* under control of a galactose-inducible promoter and does not express the gene on glucose medium. **c** Specific ATPase activity of recombinant full-length and C-terminally truncated AHA proteins assayed between pH 3.5 and pH 8.5. **d** Specific ATPase activity of recombinant full-length and C-terminally truncated AHA proteins assayed at pH 6.5 and with various ATP concentrations.

germinated later and grew slower than those of the wild-type but grew almost as long. This suggests that AHA6 and AHA9 together function in pollen tube germination and in the early phases of tube elongation, whereas AHA6 and AHA8 together are important for sustained pollen tube growth. All reported phenotypes could be functionally complemented by transforming the absent *AHA* isoforms under their native promoters back into the mutant lines (Supplementary Fig. 4). Male transmission of the fertility defects was confirmed by reciprocal crosses between the WT and plants bearing *aha* alleles (Supplementary Table 2).

Aiming to further reduce the effects of redundancy in AHA function, we generated *aha6*, *aha8*, and *aha9* triple mutants (Table 1). Pollen maturity and viability were confirmed with DAPI and Alexander stain, respectively (Supplementary Fig. 5). Pollen tube growth *in planta* was strongly affected, with only a few pollen tubes succeeding to grow through the stigma into the ovary, resulting in a reduction to about 3–10 seeds per plant (Fig. 2d–f). Likewise, in vitro germination of the triple mutant

pollen was significantly reduced (Fig. 2g), with erratic or wavy pollen tubes forming that were significantly shorter than those of the wild-type (Fig. 2i).

Although *aha6/8/9* exhibited premature pollen tube growth arrest, initial growth still occurred but at a reduced rate (Fig. 2h, Supplementary Fig. 6). A likely explanation is that one or more additional AHA isoforms is involved in pollen tube growth during the early growth phases. *AHA7* is also expressed in pollen tubes[22], but at a much lower level than *AHA6*, *AHA8*, and *AHA9* (Supplementary Fig. 1). Loss of *AHA7* led to a 30% reduction in seed set, but only when *AHA6* and *AHA9* were also lacking (Supplementary Fig. 7).

Even after genotyping >500 progeny of triple mutant plants that were pollinated with pollen from plants that were heterozygous for the fourth allele, *aha6/7/8/9* quadruple mutants were not identifiable (Supplementary Fig. 7). This suggests that *AHA7* functions redundantly with the other isoforms, and may weakly sustain pollen tube growth in the *aha6/8/9* mutant.

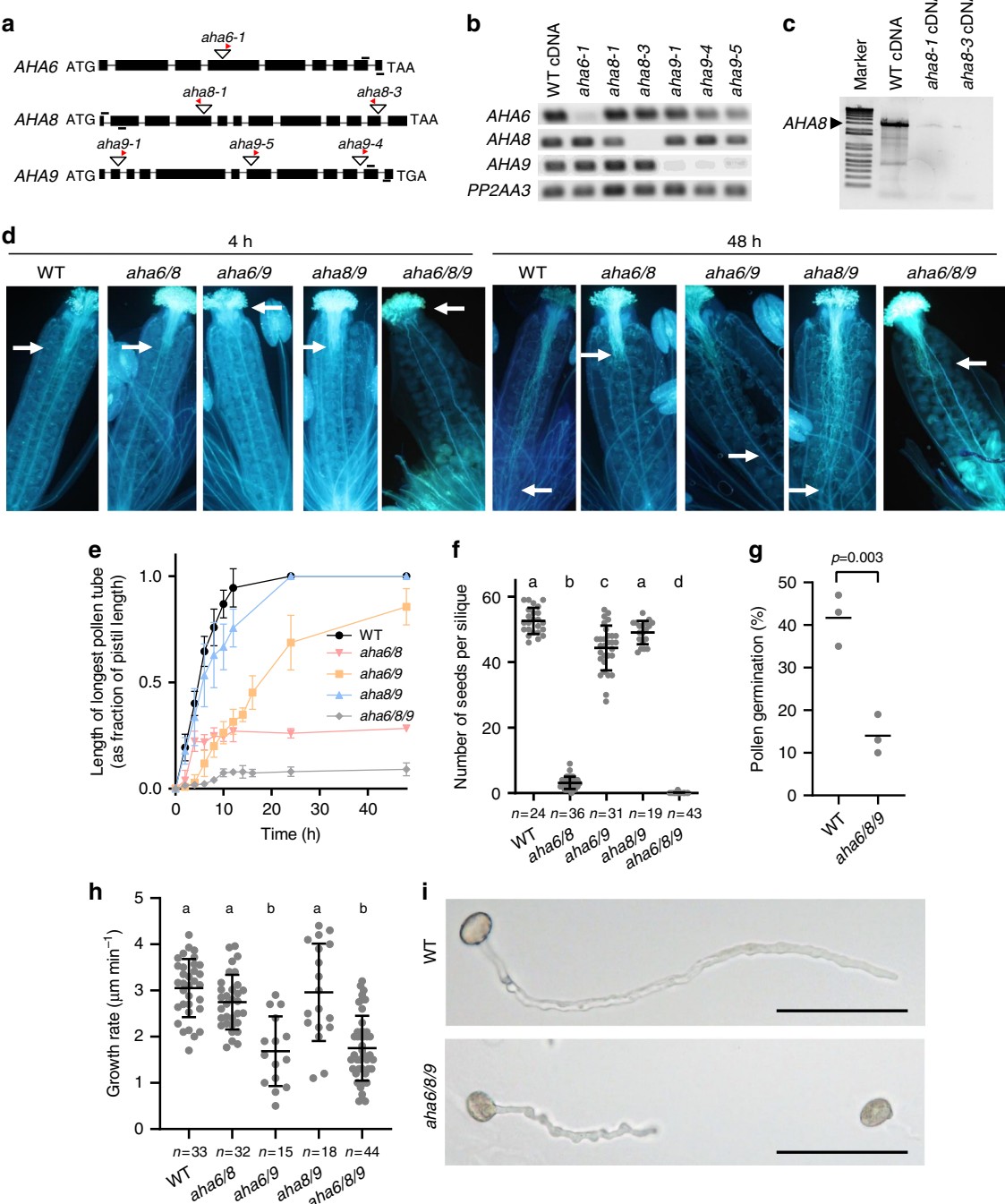

**Fig. 2 AHA double and triple mutants display aberrant pollen tube phenotypes. a** Diagram of gene structures with position of T-DNA insertions (triangles) in the genes. Red arrowheads indicate the T-DNA left border. Exons are indicated as black rectangles, and introns as black lines. Primers used for reverse-transcription PCR (RT-PCR) are indicated by short black lines above the gene representations. **b** RT-PCR analysis of T-DNA insertion lines using primer pairs indicated in (**a**). RNA was from flowers at stage 12[71]. *PP2AA3* served as a positive control. **c** RT-PCR analysis of T-DNA insertion lines using primers that amplify full-length cDNA of *AHA8*. **d** Aniline blue-stained pollen tubes of *ms1* pistils pollinated with different *aha* mutant lines. White arrows indicate the position of the longest tubes at 48 h after pollination. **e** In vivo pollen tube growth rate (as a fraction of pollen tube length over pistil length), as revealed by aniline blue staining ($n \geq 3$ for all genotypes and time points; error bars show SD). **f** Number of seeds per silique (one-way ANOVA and Bonferroni's Multiple Comparison Test; $p < 0.01$; error bars show mean with SD). **g** In vitro germination rate for WT and *aha6/8/9* triple mutant pollen. Means from three individual experiments with $\geq 100$ scored pollen grains for each genotype and replicate are shown (two-sided *t*-test; $p < 0.003$). **h** Average pollen tube growth rate assayed in vitro (one-way ANOVA; Bonferroni's test; $p < 0.01$; error bars show mean with SD). **i** Light microscopy images of WT and *aha6/8/9* pollen tubes grown in vivo. The triple mutant is characterized by short and wavy pollen tubes. Scale bar = 100 μm.

**Table 1 Offspring of self-pollinated lines exhibit a non-Mendelian pattern of inheritance.**

| Homozygous background (parent) | Heterozygous allele (parent) | AHA/AHA (progeny[a]) | AHA/aha (progeny[a]) | aha/aha (progeny[a]) | Ratio | $\chi^2$ [b] | P value[b] |
|---|---|---|---|---|---|---|---|
| aha6-1 | AHA8/aha8-1 | 46 | 38 | 2 | 1:0.8:0 | **46.19** | **<0.0001** |
| aha6-1 | AHA9/aha9-4 | 91 | 116 | 19 | 1:1.3:0.2 | **46.04** | **<0.0001** |
| aha8-1 | AHA6/aha6-1 | 65 | 60 | 5 | 1:0.9:0.1 | **56.15** | **<0.0001** |
| aha8-1 | AHA9/aha9-4 | 18 | 42 | 21 | 1:2.3:1.2 | 0.33 | 0.847 |
| aha9-4 | AHA6/aha6-1 | 78 | 85 | 16 | 1:1.1:0.2 | **43.40** | **<0.0001** |
| aha9-4 | AHA8/aha8-1 | 18 | 50 | 19 | 1:2.8:1.1 | 1.97 | 0.374 |
| aha6-1/aha8-1 | AHA9/aha9-4 | 175 | 209 | 36 | 1:1.2:0.2 | **92.01** | **<0.0001** |
| aha6-1/aha9-4 | AHA8/aha8-1 | 46 | 39 | 0 | 1:1:0 | **50.37** | **<0.0001** |
| aha8-1/aha9-4 | AHA6/aha6-1 | 41 | 35 | 0 | 1:0.9:0 | **44.71** | **<0.0001** |

[a]Alleles were detected in T1 seeds by PCR. The expected ratio for non-compromised gametes is 1:2:1.
[b]Bold font highlights significant differences.

Another possible candidate is *AHA3*, which is expressed in microspores[27]. However, this possibility could not be studied, since *AHA3* deletion mutants failed to produce pollen[28]. Together, our data suggest that AHA6, AHA7, AHA8, and AHA9 are needed for proper pollen tube germination and elongation and that these pumps function redundantly in the pollen tube.

**Ion fluxes are reduced throughout *aha6/8/9* pollen tubes**. Having established the genetic basis of AHA function in pollen tube growth, we focused on its mechanistic basis in cell physiology. We used an extracellular ion-selective vibrating probe to measure $H^+$ flux profiles along the pollen tube length, as these profiles can be taken as reporters of AHA activity[17] (representative wild-type pollen tube in Fig. 3a). The growth rate reduction in all mutant combinations lacking AHA6 (Fig. 2e, h) was associated with reduced $H^+$ influx at the tip (Fig. 3b), lower efflux at the shank, and a retraction of the influx/efflux reversal boundary towards the shank (Fig. 3c), with more dramatic effects in the triple mutant. Wild-type pollen tubes showed an inversion of net $H^+$ influx at the tip to a net efflux at a distance of ca. 15–20 μm from the tip (Fig. 3c), while all mutant combinations lacking AHA6 have virtually no net efflux along the tube (Fig. 3c). Although *aha8/9* shows an inversion point beyond 20 μm from the tip (Fig. 3c), the influx at the tip and efflux along the shank are comparable to those of the wild-type.

Large anionic fluxes were hypothesized to function in water movement, turgor, and elongation in growing pollen tubes[29], with efflux at the apex and influx along the tube and a corresponding negative cytosolic concentration gradient at the tip[30–32]. Indeed, a striking reduction of anionic efflux was observed in the double and triple mutants lacking *AHA6*, while *aha8/9* showed wild-type-like fluxes (Fig. 3d), in agreement with the growth phenotype observed in vivo and in vitro (Fig. 2e, h). Anionic efflux at the tip, which contributes to the largest electrical currents in growing lily (*Lilium longiflorum*) pollen tubes[33], appears to respond to the electrochemical imbalance generated by AHAs at the shank, which would be expected to contribute to a net hyperpolarization of the plasma membrane[32,34]. Indeed, cationic influx at the tip can be elicited by hyperpolarizing pulses applied to the shank of lily pollen tubes[32], supporting a link between tip fluxes and shank membrane potential.

**A cytosolic gradient of $H^+$ is reduced in *aha6/8/9* tubes**. Since mutants lacking *AHA6* virtually lacked $H^+$ efflux at the shank of the pollen tube, while retaining some, albeit a reduced, influx at the tip (Fig. 3b, c), we expected their pollen tubes to have a higher cytosolic $H^+$ concentration and a smaller gradient between the tip and shank. We assessed cytosolic $H^+$ concentrations by transforming all mutant lines with the ratiometric probe pHluorin, which was calibrated using the $H^+$ ionophore nigericin and medium with pH 5.8–8.0 (Supplementary Fig. 8a). The pH gradients were estimated using a custom protocol[35].

*A. thaliana* wild-type pollen tubes had a cytosolic $[H^+]$ gradient, with a higher pH in the shank than in the tip region, in agreement with the localization pattern of the $H^+$-exporting AHA6, AHA8, and AHA9 pumps (Fig. 3a). Accordingly, a lower cytosolic pH was observed at the tip and shank in all mutant combinations lacking AHA6 than in the wild-type (Fig. 3e). In the apex, the pH of the *aha6/8/9* pollen tubes was *ca.* 0.3 units lower than that of the wild-type (Fig. 3e, Supplementary Fig. 8b and exemplified in Fig. 4a). Even more pronounced differences were observed in the shank, where the pH of the mutant was *ca.* 0.5 units lower than that of the wild-type (Fig. 3e; Supplementary Fig. 8c; Fig. 4a). These changes in cytosolic pH translate into a shallower tip-to-shank pH gradient (Fig. 3f, detailed in Fig. 4a). The reduction in growth rate observed in AHA mutants is thus associated with the difference in pH and the gradient itself, suggesting a mechanistic relationship between growth and cytosolic pH. However, only the tip/shank gradient showed a significant correlation with average growth rate both across (Fig. 3g) and within the time series (Supplementary Fig. 9), while absolute pH at the tip did not (Supplementary Fig. 9).

Tip pH is considered to be involved in growth regulation, since $H^+$ influx into the tip, but not the shank, can change the directionality of pollen tube growth as previously shown in tobacco. Application of the cation ionophore gramicidin A to the apex, where NtAHA1 is absent, resulted in cytosol acidification concomitant with reorientation of pollen tube growth, but had no effect when applied to the shank, where NtAHA1 is present[17]. Similarly, growing lily pollen tubes subjected to an altered pH regime at the tip induced by water electrolysis on an active electrode either bent away from the electrode, stopped growing, or bursted[36]. Ion gradients have been implicated in multiple roles directly related to growth regulation, such as orienting actin polymerization and consequently cytoplasmic streaming, and vesicle/organellar dynamics[34]. Oriented vesicle movement has been hypothesized to occur spontaneously in pollen tubes by three basic mechanisms: (i) up a concentration gradient, (ii) towards the most depolarized plasma membrane domains, or (iii) along with osmotically driven fluid motion[34]. Changes in gradient steepness are expected to affect all such processes, thus possibly accounting for the reduction in average growth rate found in mutants lacking *AHA6*. These results indicate that AHAs are key regulators of the mean level and gradient of cytosolic $H^+$ concentration.

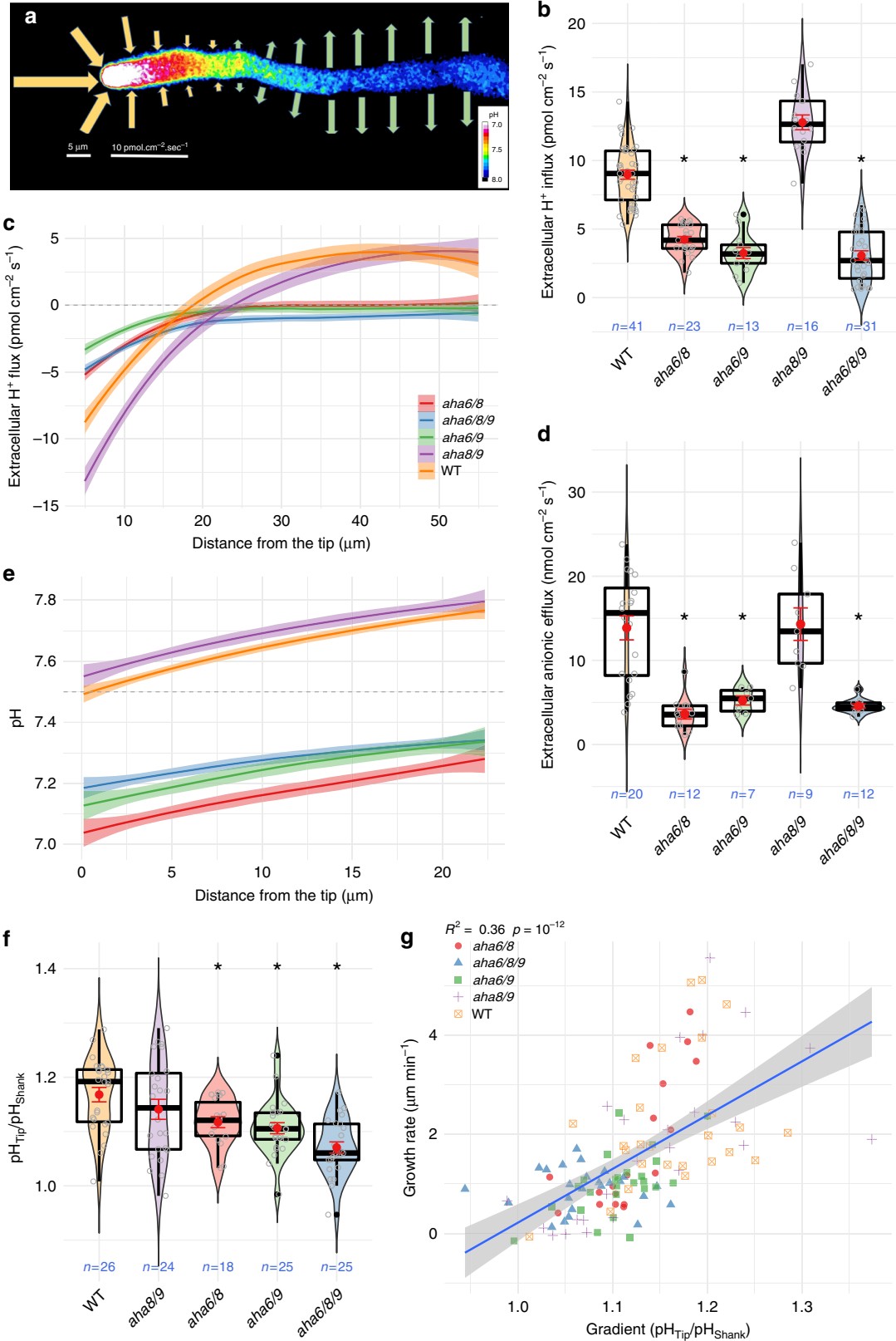

**Mutant tubes have a lower frequency of $H^+$ oscillations**. When cytosolic pH was quantified as a function of time using CHU-KNORRIS[37], we observed pronounced pH oscillations in all genotypes (Fig. 4b, c, Supplementary Fig. 10). The *aha6/8/9* triple mutant showed the largest difference from the wild-type, with an increased period of synchronized oscillations between

the growth rate and $H^+$ cytosolic concentration $[H^+]_{cyt}$ (Fig. 4d). The triple mutant is also more irregular in terms of variability in oscillation period between different pollen tubes (Supplementary Fig. 11a, b), changes in phase relationship (Fig. 4e), and highly variable delays between growth and $[H^+]_{cyt}$ (Supplementary Fig. 11c).

**Fig. 3 Reduced pollen tube growth in *aha6* double and triple mutants is associated with reduced extracellular ion fluxes and intracellular pH gradients.**
**a** Representative WT pollen tube summarizing $H^+$ fluxes measured at the surface (arrows) and cytosolic pH gradient (false color). Arrow size is scaled with the flux intensity shown on the bottom bar, while direction denotes influx or efflux. **b** Extracellular $H^+$ fluxes at the pollen tube tip. Violin plots show the probability density with color-filled curves obtained from individual observations (open gray circles), with boxplots (thick black lines and outliers as black dots) overlaid with the mean and standard error (red circle and lines). **c** Extracellular $H^+$ fluxes throughout the pollen tube sampled every 5 μm, averaged and interpolated with a local polynomial fit (loess) with $n > 10$ for all genotypes. Negative values indicate influx and positive values efflux.
**d** Extracellular anion efflux at the tip. **e** Cytosolic pH throughout the tube averaged with loess for each genotype ($n > 16$), obtained by fluorescence imaging of a calibrated pH probe (pHluorin). Media pH indicated by dashed line. **f** Gradient steepness estimated by the fold change in cytoplasmic pH between the tip and shank (subcellular regions defined as in Fig. 4). **g** Average growth rate correlates with pH ratio between the tip and shank (linear fit in blue with standard deviation shaded), while its does not correlate with tip pH alone (Fig. S9). Where applicable, asterisks indicate a significant difference ($p < 0.01$) compared to the control (WT); one-way ANOVA followed by the post-hoc Dunnett test.

All genotypes showed complex synchronized oscillations between $[H^+]_{cyt}$ and growth rate, with multiple frequencies co-existing in the same sequence, but with drastic changes in period over time (Supplementary Fig. 10), although with no overall difference in amplitude (Supplementary Fig. 11d). Slower oscillations could contribute to the drastically premature growth arrest of AHA6 mutants (Fig. 2d, e), presumably through critical effects on time-sensitive processes such as actin polymerization and cell wall rigidity[7]. AHAs could affect tip oscillations through the electrochemical imbalance generated in the shank plasma membrane, leading to hyperpolarization in the shank and a more depolarized tip[6,32,34], or via pH/voltage sensitive channels localized throughout the pollen tube, such as the recently characterized $H^+/Cl^-$ co-transporter TMEM16[38].

**Turgor pressure is not altered in mutant pollen tubes**. After having identified spatio-temporal changes in intracellular pH and extracellular ion fluxes in AHA mutants, we investigated its possible effect on the mechanism underlying the observed defect in pollen tube growth. The plasma membrane $H^+$-ATPase has been proposed to be the powerhouse driving pollen tube growth[13,14]. Plasma membrane $H^+$-ATPases energize the plasma membrane and drive the uptake of solutes through $H^+$-coupled co-transporters and channel proteins, followed by the uptake of water, which results in an increase in turgor pressure. The reduced ion fluxes in the mutant pollen tubes could thus lead to a decrease in turgor pressure, which could underlie the observed elongation defects. To test this hypothesis, we estimated turgor in growing pollen tubes using the incipient plasmolysis technique[39,40]. Despite severe tube growth phenotypes in *aha6/8/9* (Fig. 2e, i), turgor pressure in wild-type and mutant pollen tubes was not significantly different, with an average of ~0.2 MPa (Fig. 5a), which is in agreement with the results of previous studies[38,39].

The lack of correlation between internal turgor pressure and growth rate of pollen tubes[39] (reviewed in refs. [40,41]) contradicts the assumption that turgor is modulated during oscillatory growth[42,43]. Our results support a tip growth model, which is not dependent on increased turgor pressure, and suggest that the combined fluxes of the opposing circuits of cations and anions at the tip and shank balance each other out, so that there is no net build-up of osmotically relevant ionic concentrations[34].

**Actin cytoskeleton organization at the tip is compromised**. We then investigated the actin cytoskeleton as a possible target of the pH alterations that could explain the aberrant growth of *aha6/8/9* pollen tubes (Fig. 2i). Actin is essential for pollen tube growth and is organized in bundles along the shank of the tube, ending in a basket- or cortical fringe-like structure in the sub-apical region (~12–20 μm from the tip in lily, likely ~3–6 μm in Arabidopsis) and as a meshwork of very short, highly dynamic thin

microfilaments in the apical dome[44,45]. The sub-apical structures are fundamental for tip growth. Actin bundles along the shank end at this location where they are reorganized to sustain the inverted-fountain streaming pattern, with cortical motion towards the tip, and central motion towards the grain. The molecular mechanisms underlying this actin remodeling mechanism and, fundamentally, its spatial definition, remain poorly understood. Yet, pH is an established regulator of the ADF/cofilin family of actin remodeling proteins[46–48], which drives the turnover and depolymerization of entangled F-actin[49] and has been shown to function in actin organization in pollen tubes[44,50].

We thus explored the role of pH in regulating actin turnover in pollen tubes, by staining actin with Alexa 488 phalloidin[51,52]. In addition to lacking a sub-apical actin fringe, the volume of the apical 5 μm of *aha6/8/9* mutant pollen tubes was devoid of actin (Fig. 5b). Given the alterations in cytosolic pH and the tip-focused $H^+$ gradient in the absence of AHAs, the changes in actin distribution are likely caused by differences in the threshold of this tip-focused gradient. This interpretation is further in accordance with the cytosol acidification-induced disruption of actin filament organization in lily pollen tubes[44].

On the other hand, cytosol alkalinization promoted by plasma membrane $H^+$-ATPase was proposed to stimulate ADF, leading to depolymerization of F-actin and more plus ends, which then stimulate actin filament polymerization and, eventually, growth[44]. Interestingly, *aha6/8/9* phenocopies mutants in actin turnover factors that also display disrupted apical actin fringes and wavy growth, such as *adf10* (normally present throughout the tube)[53], CAP1 (which acts in synergy with the ADF/cofilin family)[54–56], and *A. thaliana* formin homology3 (AtFH3) and AtFH5[57]. A shallower pH gradient (Fig. 3e, f) could also impact the orientation of actin fibers, since a gradient in pH should result in a gradient of ADF activity, which is pH dependent[46–48] and hence could spatially define domains of re-organization and de novo polymerization.

Together, these results suggest that the abnormal growth and erratic shape (Fig. 2i) of AHA mutant pollen tubes are caused by a defect in actin turnover at the pollen tube tip (Fig. 5b), induced by reductions in cytosolic pH, pH gradient, and frequency of $H^+$ oscillations (Fig. 4 and Supplementary Fig. 11).

**Membrane potentials are less hyperpolarized in *aha* mutants**. AHAs are electrogenic pumps that establish a pH gradient across the membrane by pumping $H^+$ out of the cell, thus hyperpolarizing the cell to more negative values of membrane potential. For quantitative comparisons of the membrane potentials in wild-type and mutant pollen tubes, we took advantage of ANNINE-6 plus (Fig. 5d), which is a voltage-sensitive fast-response dye that has been used for sensitive optical recording of neuronal excitation[58]. Although methodological proof-of-principle demonstrations have been published[59,60], this dye has not been used for

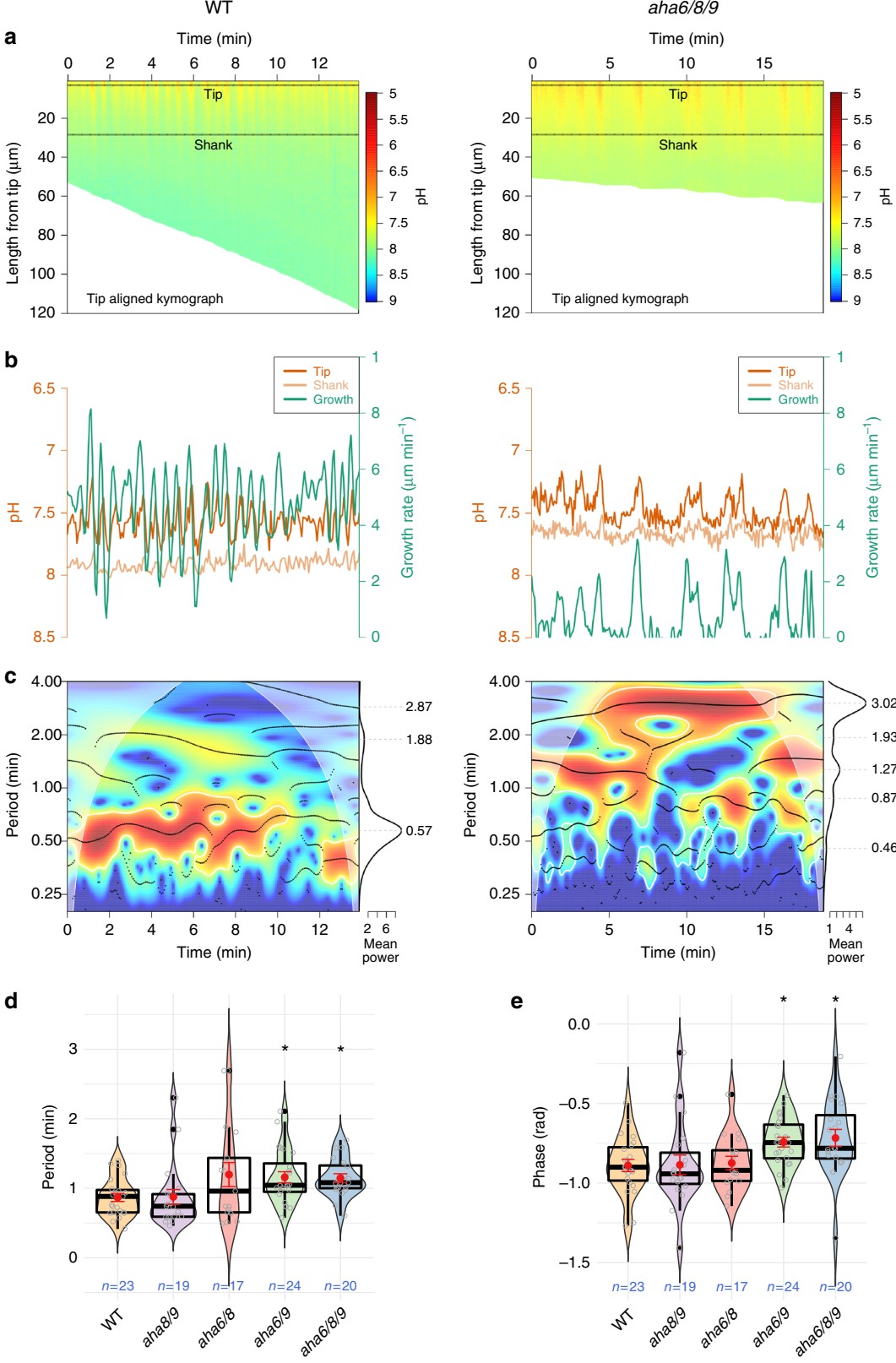

plant cell phenotyping. We focused on the sub-apical zone (10–20 μm distal from the tip, where AHA labeling fades away) and the AHA-stabilized shank (40–50 μm from the tip) (Supplementary Fig.12 a–e). From this analysis, it was evident that the *aha6/8/9* mutants had less hyperpolarized membrane potentials than the wild-type, (<50% fluorescence) (Fig. 6d). The wild-type

membrane potential has previously been determined to be at −127 mV[61]. As the linearity of the ANNINE6-plus response at such negative values has not been determined, one cannot extrapolate exact values for the mutant, but given the linearity of the dye response over 120 mV (reviewed in ref. [58]), values more positive than −70 mV should be expected for the triple mutant.

**Fig. 4 Synchronized oscillations between cytosolic pH and growth rate are slower in *aha6/8/9* pollen tubes. a** Tip-aligned kymograph with calibrated pH values. Subcellular tip and shank regions of interest defined by the highlighted regions. **b** Growth rate (green) and cytosolic pH at the tip (orange) and shank (light orange). **c** Synchronization between pH at the tip and growth rate oscillations. Significant joint periodicity across time is shown in the cross-wavelet power spectrum (heatmap) demarcated by white lines ($p < 0.05$ compared to an autoregressive process of order 1), whereas the main instantaneous periods are represented by black dots corresponding to peaks in power (wavelet ridges). The mean power, shown at the right, is averaged across all time points outside the cone of influence (pale region where the estimation is prone to distortion). The overall mean periods present in a series are seen as peaks, which are indicated by gray dashed lines and their numerical values. Median significant period of synchronized pH/growth oscillations (**d**) and their mean phase relationship (**e**) for each series show differences across genotypes. Violin plots show the probability density with color-filled curves obtained from individual observations (open gray circles), with boxplots (thick black lines and outliers as black dots) overlaid with mean and standard error (red circle and lines). Asterisks show significance according to a one-way ANOVA followed by the post-hoc nonparametric Dunnett test ($p < 0.05$).

We further confirmed this result using the oxonol dye DiBAC$_4$(3), a slow-response voltage-sensitive dye that enters depolarized cells and is excluded from hyperpolarized cells[62]. We have assayed DiBAC$_4$(3) under normal conditions and after wortmannin treatment to prevent possible endocytosis artifacts on dye uptake. Under all conditions, the *aha6/8/9* triple mutant had an increased degree of DiBAC$_4$(3) uptake compared to the WT, which is indicative of a less negative potential (Supplementary Fig. 12f).

The quantitative accuracy of ANNINE-6 and the confirmation of its main conclusion using DiBAC$_4$(3) provide compelling evidence that pollen tubes of AHA mutants, which are impaired in growth and fertility, have less hyperpolarized membrane potentials, thus substantiating a mechanistic role for AHAs in pollen tube growth through plasma membrane energization.

This work provides the first genetic evidence that plasma membrane H$^+$-ATPases energize the plasma membrane and control dynamic H$^+$ fluxes in growing pollen tubes and are required for apical elongation of pollen tubes and hence successful fertilization. Furthermore, these proton pumps underlie the spatiotemporal control of the pH homeostasis set point, establishing a more acidic apex than shank region and a pattern of proton oscillations that is synchronized with pollen tube growth. We propose that the spatial distribution of plasma membrane H$^+$-ATPases, which are present only at the tube shank, mediates the formation of a [H$^+$]$_{cyt}$ gradient that directs actin bundling near the growing tip, and generates a proton motive force that drives the anion efflux and H$^+$ influx at the tip, producing an oscillatory ion circuit that coordinates pollen tube growth (Fig. 6). Local ionic fluxes precede pattern formation and polarized growth in a diverse array of tip-growing cells such as algal zygotes, fungal hyphae, and growing neurons (reviewed in ref. [33]), such that depolarization of the tip may be a pre-condition to sense external stimuli[5]. Indeed, ion gradients have been reported in virtually all biological models for polar growth[63], while H$^+$ has long been proposed as a crucial signal for pollen tube growth in particular[64,65]. Our results are in line with such models, supporting the notion that AHAs have multiple effects on tip growth. Furthermore, the approach presented here might help elucidate the mechanisms driving growth in other types of tip-growing cells.

## Methods
**Plant materials and growth conditions**. *Arabidopsis thaliana* ecotype Columbia (Col-0) was used as the wild-type plant. T-DNA insertion lines were obtained from the Nottingham Arabidopsis Stock Centre. Seeds were sown on soil and vernalized in darkness at 4 °C for 2 days before being transferred to climate chambers maintained at 22 °C with 70% humidity and a 16-h-light/8-h-dark cycle (100 μmol. m$^{-2}$.s$^{-1}$). For ion flux measurements, plants were grown at 22 °C with 70% humidity and a 12 h light/12 h dark cycle (100 μmol m$^{-2}$ s$^{-1}$).

**Characterization of T-DNA insertion mutants**. Genomic DNA was used to identify heterozygous and homozygous lines by PCR. Gene-specific primers

together with specific T-DNA left border primers (Supplementary Table 3) were used to genotype T-DNA insertion lines. All T-DNA insertion lines were verified by sequencing. *AHA9* transcript levels in Col-0 and the respective mutants. Total RNA was extracted from flowers (10 flowers per sample; at flower stage 12[66]) using a Plant RNeasy Kit (Qiagen) according to the manufacturer's instructions. To synthesize cDNA, 1 mg RNA was reverse transcribed using the iScript cDNA Synthesis Kit (BioRad), according to the manufacturer's instructions. RT-PCR was performed using 5 ng cDNA (or 1 ng gDNA extracted from a Col-0 plant) as template and cDNA-specific primer pairs (Supplementary Table 3) were used to determine the levels of full-length and partial transcripts. *PP2AA3* (At1g13320) was amplified as a positive control for RT-PCR analysis[67] (Supplementary Table 3).

The *aha9-1* allele had a pleiotropic phenotype for heterozygous, but not homozygous, plants, probably due to chromosomal rearrangements[68], and was thus not further characterized.

**Segregation analysis**. Plants heterozygous for a T-DNA insertion allele were allowed to self-fertilize and the resulting seeds were sown on soil in 96-cell plug trays. Genomic DNA was extracted from leaves as described below and the alleles in question were amplified by PCR using sets of optimized primer pairs (Supplementary Table 3). For reciprocal crosses, unopened flowers of plants heterozygous for a T-DNA insertion were emasculated and stigmas were hand pollinated with WT pollen. Flowers of *ms1* plants were pollinated with plants heterozygous for a T-DNA insertion. Developed seeds were collected from these siliques and sown on soil in 96-cell plug trays, and the genotypes were determined as described above.

**DNA extraction**. DNA was extracted in 96-well format. Leaves of young plants were collected in tubes each containing a 3 mm steel ball (Dansk Kugleleje Center). Samples were frozen in liquid nitrogen and homogenized using a Retsch Tissue Lyser (QIAGEN). Then, 280 μl Edward's solution (200 mM Tris-HCl [pH 7.5], 250 mM NaCl, 25 mM EDTA, and 0.5% (w/v) sodium dodecyl sulfate)[69] was added to each sample and the 96-well plates were centrifuged for 15 min at 5900 rcf at room temperature. Then, 125 μl buffer was transferred to fresh 96-well plates containing 125 μl 2-propanol and mixed by pipetting. Plates were centrifuged for 15 min at 5900 rcf. The supernatant was pipetted off and discarded, 300 μl 70% (v/v) ethanol was added to the pellet without resuspending it, and the samples were centrifuged for 5 min at 5900 rcf and room temperature to wash the pellets. The ethanol was removed and the samples were left to dry overnight at room temperature. The DNA was resuspended in 300 μl deionized water and the samples were maintained at room temperature for at least 3 h. DNA samples were diluted 10-fold in deionized water and for genotyping purposes 5 μl DNA was used per 20-μl PCR reaction.

**Construction of the *promoter::gene::GFP* fusions**. To construct the *pAHA6::AHA6::GFP*, *pAHA8::AHA8::GFP*, and *pAHA9::AHA9::GFP* expression plasmids, genomic *A. thaliana* Col-0 DNA was amplified by Phusion High-Fidelity DNA-polymerase (NEB) using primer pairs that included the region 1747 (*AHA6*), 673 (*AHA8*), or 762 (*AHA9*) bp upstream of the start codon and the entire gene except for the stop codon (Supplementary Table 3). The resulting PCR products were cloned into pENTR/D-TOPO vectors (Invitrogen) by TOPO cloning, according to the manufacturer's instructions, and the inserts were fully sequenced. Using LR recombinase (Invitrogen), the inserts were subsequently transferred into the pMDC107 destination vector, such that the C-termini of the inserts were fused in-frame to EGFP[70]. The resulting expression vectors were then introduced into Col-0 and the *aha6/8* and *aha6/9* double mutant lines.

**Construction of the *pAHA8::AHA8* construct**. To construct the *pAHA8::AHA8* expression plasmid, genomic *A. thaliana* Col-0 DNA was amplified by Phusion High-Fidelity DNA-Polymerase (NEB) using the same forward primer as used for cloning the GFP construct, and reverse primer including the stop codon (Supplementary Table 3). The resulting PCR product was cloned into pENTR/D-TOPO vectors (Invitrogen) by TOPO cloning, according to the manufacturer's instructions, and the inserts were fully sequenced. Using LR recombinase (Invitrogen), the

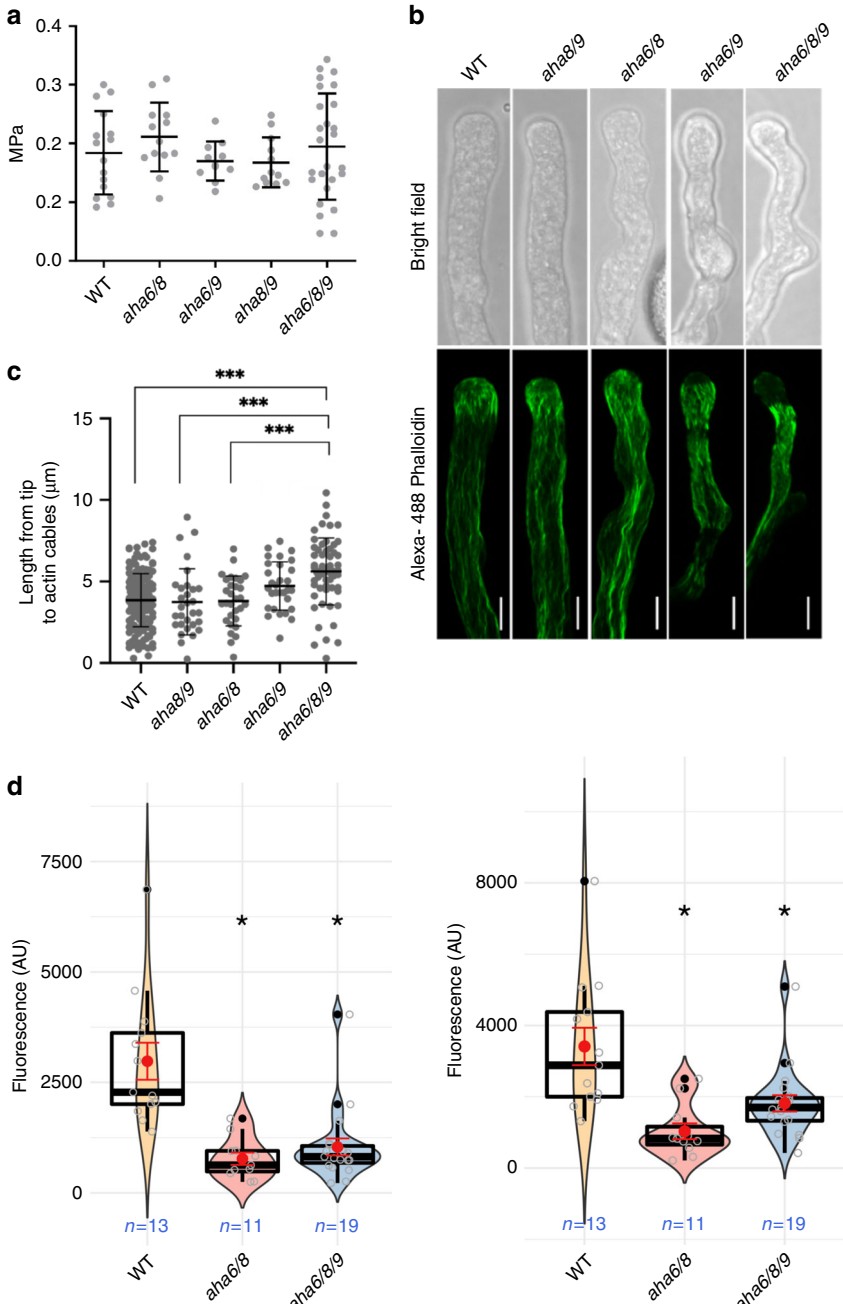

**Fig. 5 Turgor pressure is unaltered in *aha6/8/9* pollen tubes, but actin is located away from the tip and membrane potential is reduced. a** Turgor pressure in WT and AHA mutant pollen tubes (one-way ANOVA and Bonferroni's Multiple Comparison Test; $p < 0.01$; error bars show mean with SD). **b** Actin in pollen tubes was stained with Alexa 488 phalloidin and images were obtained using a confocal laser scanning microscope. Fluorescence and bright-field images were generated by maximum intensity projections of all optical sections for each pollen tube using ImageJ software (scale bars = 5 μm). **c** The distance from the tip of the pollen tube to the actin filaments in pollen tubes. Where applicable, asterisks indicate a significant difference compared to the control (WT); ANOVA with Bonferroni's test; $p < 0.01$; error bars show mean with SD. **d** Quantification of plasma membrane potential using ANNINE-6-plus 10–20 μm (left panel) and 40–50 μm (right panel) from the tip, respectively. Violin plots show the probability density with color-filled curves obtained from individual observations (open gray circles), with boxplots (thick black lines and outliers as black dots) overlaid with mean and standard error (red circle and lines). Membrane fluorescence values were quantified using a customized protocol, including near super-resolution, Nyquist oversampling and normalization between replicates (Suppl. Fig. 12). Differences were significantly lower in all mutants than in wild-type, but no significant differences were found within mutants at either domain of the plasma membrane sampled (see Suppl. Fig. 12 for details) (scale bars = 2 μm).

inserts were subsequently transferred into the pMDC99 destination vector[70]. The resulting expression vector was then introduced into the *aha6/8* double mutant line.

**Construction of the *promoter*∷*GUS* constructs.** To construct the *pAHA6::GUS*, *pAHA8::GUS*, and *pAHA9::GUS* expression plasmids, genomic *A. thaliana* Col-0

DNA was amplified by Phusion high-fidelity DNA-polymerase (NEB) using the same forward primer as used for cloning the GFP constructs, and reverse primers complementary to sequences in the second exon of the respective gene (Supplementary Table 3). The resulting PCR products were cloned into pENTR/D-TOPO vectors (Invitrogen) by TOPO cloning, according to the manufacturer's instructions, and the inserts were fully sequenced. Using LR recombinase (Invitrogen), the inserts were subsequently transferred into the pMDC162 destination vector[70].

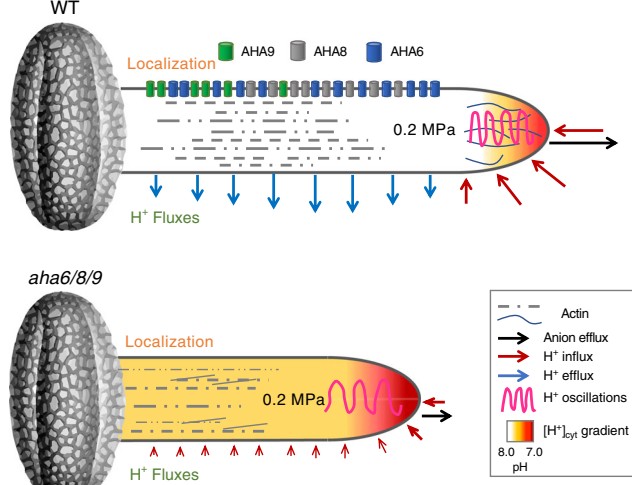

**Fig. 6 Model implicating plasma membrane H⁺-ATPases (AHAs) in the spatio-temporal control of ion fluxes and intracellular gradients required for actin organization during pollen tube growth.** In the absence of AHAs (barrel shapes), extracellular $H^+$ fluxes (colored arrows) and anionic efflux (black arrow) vanish, leading to a lower pH throughout the tube with a shallower cytosolic pH gradient (color fill heatmap), lower frequency of synchronized growth rate/$[H^+]_{cyt}$ oscillations (magenta trace), and absence of organized actin at the tip (thin black lines), ultimately resulting in growth defects (represented by a shorter tube). Modified image of an Arabidopsis pollen grain by courtesy of Prof. David Twell, Electron Microscopy Facility, College of Life Sciences, University of Leicester.

**Construction of *pLat52::pHluorin* constructs**. Ratiometric pHluorin is a pH-sensitive GFP variant[71,72], which carries a cryptic intron[72], leading to aberrant mRNA processing and the absence of protein expression. Silent mutations were introduced into the pHluorin coding sequence, using a primer pair (Supplementary Table 3), to eliminate the cryptic intron sequence and restore pHluorin expression in *A. thaliana*. The resulting amino acid sequence is 100% identical to the original pHluorin. The *aha9-4* line is resistant to kanamycin, most likely due to a kanamycin resistant marker being associated with the T-DNA inserted into *AHA9*. Therefore, the *pLat52::pHluorin* sequence was shuttled into pMDC99[70], which in plants is selected for by hygromycin B. Briefly, *pLat52::pHluorin* plasmid and pMDC99 were mixed and digested using *Xba*I and *Hin*dIII (NEB), effectively excising *pLat52::pHluorin* and the Gateway Cassette. The digestion product was ligated with T4 DNA ligase (NEB) and transformed into *Escherichia coli* strain TOP10. Positive clones containing the resulting pMP5249 vector were selected on LB plates supplemented with 50 mg ml⁻¹ kanamycin (Supplementary Fig. 13 for vector map).

**Construction of the *pKI.1 R* CRISPR/Cas9 constructs**. As *aha6/8/9* produced too few seeds to allow transformation with the *pLat52::pHluorin* construct, the *aha6/8/9::pHluorin* line was produced using CRISPR. CRISPR/Cas9-mediated gene knockout of *AHA6* or *AHA8* in the *aha6/9::pHluorin* or *aha8/9::pHluorin* lines was done using the pKAMA-ITACHI vector pKI1.1 R (Addgene)[73]. sgDNA was designed to target the first exon of either AHA6 or AHA8 using CRISPOR[74] and CRISPR-P 2.0[75]. Primers for cloning were designed to include the sgDNA sequence and the AarI restriction digestion site (Supplemental Table 3). Primers were annealed using T4 polynucleotide kinase (NEB) and the pKI1.1 R vector was digested using AarI (Thermo Scientific) before being dephosphorylated using Shrimp Alkaline Phosphatase (rSAP, NEB). Vector and hybridized primers were ligated using the T4 ligase (NEB), transformed into *E. coli* and selected on LB plates supplemented with 100 mg/L spectinomycin. Plants were transformed as described below. Transformants showing red fluorescence in the seed due to TagRFP were selected using a Leica FM Stereo Microscope M205 FA with a DsRED Filter (excitation, 546/10 nm; emission, 600/40 nm). The *aha6/8/9* triple knockout plants were selected based on their strongly reduced seed setting phenotype.

**Plant transformation**. Plasmids were transformed by electroporation into *Agrobacterium tumefaciens* strain GV3101::pMP90[76] and transformants were selected on YEP plates (1% [w/v] yeast extract, 1% [w/v] bacto peptone, 0.5% [w/v] NaCl, 1.5% [w/v] bacto agar) containing 50 mg ml⁻¹ kanamycin and 25 mg ml⁻¹ gentamycin. Destination vectors extracted from *A. tumefaciens* single colonies were sequenced to verify the inserts. *A. thaliana* plants were transformed with *A. tumefaciens* (strain GV3101::pMP90) containing the desired vectors. For plant

transformation, a modification of the floral-dip method was used[77]. Briefly, *A. tumefaciens* cells were grown on three plates containing YEP, 50 mg ml⁻¹ kanamycin, and 25 mg ml⁻¹ gentamycin for 48 h. Cells were then resuspended in 150 ml infiltration solution containing 5% (w/v) sucrose and 0.03% (v/v) Silwet L-77 (Lehle seeds). Inflorescences of newly bolted *A. thaliana* plants were dipped in the infiltration solution for 30 s and subsequently covered with a plastic bag overnight. This was repeated three times, allowing the plants to recover for 5–7 days between infiltration treatments.

**Selection of transformed plants**. Seeds of infiltrated plants were surface sterilized by washing in 70% (v/v) ethanol for 10 min, bleach solution with 0.05% (v/v) Tween-20 for 5 min, and 70% (v/v) ethanol for 5 min, and then rinsing three times in milliQ water. Seeds were plated on ½ strength Murashige-Skoog (MS) medium[78] supplemented with 1% (w/v) sucrose and 50 mg ml⁻¹ hygromycin B. The plates were stored at 4 °C for 48 h in darkness, and then transferred to growth chambers at 20 °C with a 16-h light period. After 2 weeks, seedlings showing resistance to hygromycin B were transferred to soil and tested by PCR for the presence of the correct insert.

**Isolation of pollen**. Unless stated otherwise, pollen grains were isolated in the morning from newly opened flowers. For this, flowers occupying a volume of about 500 µl were collected in a microcentrifuge tube and 700 µl pollen germination medium (5 mM CaCl₂, 5 mM KCl, 1 mM MgSO₄, 0.01% [w/v] H₃BO₃, 15% [w/v] sucrose, pH 7.5) was added to each tube. The samples were vortexed for 2 min, and the pollen pelleted by centrifugation at 11,200g for 2 min. Flowers and medium were removed and pollen were resuspended in fresh pollen germination medium.

**GUS staining and analysis**. To determine the tissue-specific expression of *AHA6*, *AHA8*, and *AHA9*, plants for each GUS-expressing line were selected and inflorescences were incubated for 5 min for GUS activity. Samples were vacuum infiltrated for 5 min with 5-bromo-4-chloro-3-indolyl β-D-glucuronic acid (X-glca; Duchefa) solution (3 mM K₃[Fe(CN)₆], 3 mM K₄[Fe(CN)₆], 0.4% [w/v] Tween-20, 50 mM KH₂PO₄/K₂HPO₄, pH 7.2, 2 mM X-glca) and incubated in the solution at 37 °C in darkness for 6 h (inflorescences). Pollen were incubated for 3 h at 22 °C before staining for GUS activity at 22 °C for 20 h. After incubation, inflorescences were cleared in 70% (v/v) ethanol overnight. Pollen were imaged directly. Stained samples were observed with a Leica DMR HC light microscope (pollen) and a Canon ELPH100HS camera (inflorescences).

**Subcellular localization studies**. The subcellular localization of AHA6, AHA8, and AHA9 tagged with EGFP in mature pollen and pollen tubes from the *A. thaliana* complementation lines was determined by confocal microscopy. Images were obtained with a Leica SP5, using a 40× oil immersion objective (NA 1.0; Leica). EGFP was excited with the 488-nm line of an argon laser and emission recorded from 495 to 550 nm. Optical sections were captured, and images were prepared by generating max-intensity projections (using ImageJ) of the optical sections through an individual pollen grain or pollen tube. Autofluorescence (monitored in wild-type mature pollen and pollen tubes) was negligible relative to EGFP fluorescence.

**Full-length and C-terminally truncated cDNA constructs**. Total RNA was extracted from *A. thaliana* Col-0 using an RNeasy Plant Mini Kit (Qiagen) and RNA was reverse transcribed using the iScript cDNA Synthesis Kit (BioRad) according to the manufacturers' instructions. The PCR products of full-length coding sequences and C-terminally truncated sequences (see Supplemental Table 3 for primer sequences) were cloned into the pENTR/D-TOPO vectors (Invitrogen) by TOPO cloning, according to the manufacturer's instructions, and the inserts were fully sequenced. Using LR recombinase (Invitrogen), the inserts were subsequently transferred into the destination vector pMP4409. pMP4409 was generated by cutting pMP625 containing the promoter and terminator of *PMA1* (yeast endogenous plasma membrane H⁺-ATPase[79]) with *Xho*I and *Spe*I, removing *AHA2*, and inserting the gene into Gateway Cassette A (Invitrogen); the correct orientation of the Gateway Cassette A was verified by sequencing.

**Yeast complementation assays**. AHA6, AHA8, and AHA9 expression clones were transformed into *Saccharomyces cerevisiae* strain RS-72 (*Mat a; ade1-100 his4-519 leu2-3 leu112 pPMA1-pGAL1*) to be used for complementation tests[80]. In RS-72 the natural constitutive promoter of the endogenous yeast plasma membrane H⁺-ATPase *PMA1* is replaced by the galactose-dependent promoter of *GAL1*.Since *PMA1* is essential for yeast growth, RS-72 grows only on galactose medium. Using this strain, plasmid-borne plant H⁺-ATPases brought under control of the constitutive *PMA1* promoter can be tested for their ability to rescue *pma1* mutants on glucose medium. Yeast was grown for 3 days at 30 °C in liquid medium containing 2% galactose. Each complementation experiment was replicated independently three times. In drop tests, cells were diluted in sterile water and 5 µl was spotted on selective media. Growth was recorded after incubation for 3 days at 30 °C.

**Protein expression, purification, and characterization**. Microsomes harboring recombinant plasma membrane H$^+$-ATPase were isolated from transformed yeast cells[81]. In brief, plasmids harboring the truncated AHA6, AHA8 and AHA9 was transformed into the RS-72 strain and transformed yeast strains were grown on minimal media w/o leucine. Positive clones were grown for two days in minimal medium with 2% galactose. Cells were transferred to YPD media for 20 h and harvested by centrifugation at 4.000$g$ for 10 min. The cell pellets were washed with cold MilliQ water twice and resuspended in a buffer containing 20% glycerol, 50 mM MES/KOH (pH 6.5), 10 mM EDTA, 1 mM dithiothreitol (DTT) and 0.4 mM phenylmethylsulfonyl fluoride. Cells were lysed by vortexing with glass beads (0.5 mm in diameter) 5 times 1 min. The cell debris was removed by centrifugation at 10.000$g$ for 15 min followed by an ultra-centrifuge spin at 200.000$g$ for 1 h to pellet the total membranes. The membrane pellet was washed with a buffer containing 20% glycerol, 50 mM MES/KOH (pH 6.5), 1 mM EDTA, 1 mM DTT and 0.4 mM phenylmethylsulfonyl fluoride, centrifuged again and homogenized in the same buffer. The protein concentration was determined using the Bradford reagent and bovine serum albumin as standard. The resuspended membrane fractions were either assayed directly or flash frozen in liquid nitrogen and stored at −80 °C until use. The ATPase activity was determined by assaying the liberation of phosphate according to the Baginski assay[82] in microtiter plates with a total volume of 60 µL in a buffer containing 50 mM MOPS/MES, 8 mM MgSO$_4$, 5 mM NaN$_3$, 0.25 mM NaMoO$_4$, 25 mM KNO$_3$, 2 mM phosphoenolpyruvate, and 30 µg/mL pyruvate kinase from rabbit muscle (Sigma-Aldrich). The assay buffers were equilibrated to 30 °C, and the assays were started by adding ATP and 150 ng isolated microsomes. The ATPase activity was measured in a pH range 5.4–8.1 and with 3 mM ATP to determine the pH optima. For kinetic characterization of the ATPase activity, the pH was constant at 6.5 and 12 ATP concentrations were tested in the range 0–3 mM. All experiments were performed in triplicate with ± SE.

**Aniline blue staining**. Pollen tubes in pistils were stained with Aniline blue[83]. Briefly, unpollinated *ms1* pistils (NASC ID: N75)[84,85] were pollinated either with wild-type or mutant pollen. After the indicated period, flowers with pollinated pistils were collected, vacuum infiltrated with fixing solution containing ethanol: acetic acid (3:1, v/v), and incubated overnight at room temperature. The fixed flowers were rehydrated stepwise (10 min each) with 70%, 50%, and 30% ethanol, and finally milliQ water, followed by incubation in softening solution (8 M NaOH) overnight. The softening solution was removed, and pistils were stained in 2 mL aniline blue solution (0.05% (w/v) aniline blue in 0.1 M PSB buffer, pH 9) with 2% (v/v) glycerol overnight in the absence of light and at room temperature.

Pollen tubes inside pistils were observed using a Leica DMR HC fluorescence light microscope with a ×5 objective and filter cube A (excitation filter BP 340/380, beam splitter at 400 nm, emission filter LP 425) (Leica Microsystems). Pollen tube lengths were measured using ImageJ software (https://imagej.nih.gov/ij/).

**DAPI stain for pollen development**. Ten newly opened flowers were collected in a microcentrifuge tube and 750 µL pollen isolation buffer (PIB) (100 mM NaPO$_4$, pH 7.5, 1 mM EDTA, 0.1% [v/v] Triton X-100) was added[86]. Samples were vortexed for 1 min and centrifuged for 30 s at 1,500 rcf and room temperature. Flowers and supernatant were carefully removed. Pollen were resuspended in 50 µL of 2.5 µg/ml DAPI (4′,6-diamidino-2-phenylindole) stain in PIB buffer and incubated at room temperature for 15 min. Images were acquired with a Leica SP5-X confocal microscope (×63 water objective; NA 1.2; 355 nm excitation; emission recorded between 419 and 500 nm).

**Alexander stain for pollen viability**. About 20 newly opened flowers were collected in a microcentrifuge tube and 1 mL of simplified Alexander stain[87] was added. Samples were vortexed for 2 min to release pollen, and then centrifuged for 2 min at 12,000 rcf and room temperature. Flowers and supernatant were carefully removed. Pollen were resuspended in residual supernatant and assayed with a Leica DMR HC light microscope.

**Seed setting assays**. The fertility of wild-type, mutant, and complemented plants was evaluated by counting the number of seeds in mature siliques produced on the main stem. Siliques were cleared in 70% (v/v) ethanol and images of cleared siliques were captured with a Leica DM5000B light microscope using a ×1.25 objective.

**Extracellular proton and anionic flux measurements**. An ion-selective vibrating probe (VP) was used to estimate extracellular anionic and proton fluxes at the tip and shank of pollen tubes germinated in vitro[17,30]. For that, wild-type and mutant pollen grains were collected from fresh flowers grown under short-day conditions and then germinated in liquid medium containing 500 µM KCl, 500 µM CaCl$_2$, 125 µM MgSO$_4$, 0.005% (w/v) H$_3$BO$_3$, 125 µM HEPES, and 16% (w/v) sucrose at pH 7.5[88]. Pollen grains were incubated at 21.5 °C for at least 3 h and then growing pollen tubes with a length of ≥150 µm were measured. To build the ion specific electrodes, we used the Hydrogen Ionophore Cocktail B (number 95293 Sigma) for proton fluxes measurements, and the Cl$^-$-selective liquid exchange cocktail (number 24899 Sigma) to measure anionic fluxes. As the chloride ionophore detected other anions in addition to Cl$^-$, the estimates were referred to anionic fluxes and not chloride

fluxes. All measurements were obtained as close as possible to the plasma membrane without touching the pollen tube. Fluxes were calculated using Fick's law and Cl$^-$ (−2.03 × 10$^{-5}$ cm$^2$ s$^{-1}$) and proton (9.37 × 10$^{-5}$ cm$^2$ s$^{-1}$) diffusion coefficient in aqueous solution at 25 °C. Simultaneous widefield imaging of pollen tubes was performed on a custom-made device using an inverted Nikon Eclipse TE300 equipped with an Andor iXon3 camera on the bottom Kohler port, and a Lumen 200Pro Fluorescence Illumination System (Prior Scientific).

**Ratiometric [H$^+$]$_{cyt}$ imaging and pHluorin calibration**. Transgenic wild-type pollen tubes expressing the modified ratiometric pHluorin driven by the *LAT52* promoter[89] were imaged every 4 s on an Applied Precision Deltavision Core system, mounted on an Olympus inverted microscope, equipped with a front-illuminated sCMOS camera (2560 × 2160, pixel size 6.45 µm), and an InsightSSI fluorescence illuminator, using a ×63 1.2NA water immersion objective. Filter sets were as follows: excitation 390/18 nm (DAPI) and 475/28 nm (FITC); emission 435/48 nm (DAPI) and 523/36 nm (FITC). To assess cytosolic pH in pollen tubes, ratiometric pHluorin was calibrated using 140 mM KCl, 30 µM nigericin (Tocris Bioscience), and 10 mM sodium phosphate with molar ratios of mono- and dibasic forms appropriate for the given pH (5.8–8.0). Ratiometric images of wild-type and mutant pollen tubes were obtained with the same acquisition protocol, and the calibration curve fitted with linear regression was applied to estimate the intracellular pH in all lines analyzed. Kymographs were generated from time-lapse images using ImageJ (Multiple Kymograph plugin) averaging over a 7-pixel neighborhood along a manual trace through the pollen tube midline for both DAPI and FITC channels.

**Time series and gradient analysis**. The CHUKNORRIS algorithm was used to analyze and extract oscillatory parameters from ratiometric kymographs[36]. CHUKNORRIS detects the pollen tube tip with subpixel resolution in each time point of the kymographs by fitting a linear model to the fluorescence decay of the cytosolic fluorescence signal towards the background. Tip detection allows aligning kymographs by the apex and extracting growth and fluorescence time series throughout the tube. Each time series is detrended and smoothed using a discrete wavelet transform, which separates changes in baseline (trend) and noise from the periodic signal that is then further analyzed. Wavelet transforms are used to provide time-explicit estimates of period and amplitude for individual series, additionally with phase relationships and delays for pairs. Synchronization oscillations between tip pH and growth rate oscillations were compared across genotypes using averages of the significant periodic components detected with a cross-wavelet transform for each pair of series. Although periods are often distributed in multiples modes, extracting only the main mode yielded similar results. Gradients throughout the pollen tube were analyzed by averaging a tip-aligned kymograph with a local polynomial fit (loess) across all time points within the same growth regime (protocol detailed in ref. [34]).

**Turgor pressure measurement**. The turgor pressure of pollen tubes was measured using the incipient plasmolysis method[39]. Pollen tubes were germinated for 3 h in liquid pollen germination medium in glass bottom dishes. After the germination period, 60 µl of growth medium was collected for osmolality measurement (denoting πe). Pollen tubes were observed under an inverted bright-field microscope (Leica inverted DMI4000B), and one growing pollen tube was selected for analysis. Liquid pollen germination medium with 10% (w/v) sucrose and 10% (w/v) mannitol was added to the growth medium until the plasma membrane retracted from the cell wall of the tube tip (πi). An aliquot of 60 µl of the solution was collected for osmolality measurement denoting πi. The osmolality was measured using an osmometer (Gonotec). Turgor pressure is given by πi − πe and the measured values in osmol*kg$^{-1}$ were converted to MPa at 25 °C.

**Alexa 488 phalloidin actin staining**. Pollen tubes were germinated and grown on solid pollen growth medium for 5 h at 23 °C. For actin staining[51,90], the pollen tubes were fixed in fixation buffer (300 µM 3-maleimidobenzoic acid *N*-hydroxysuccinimide ester in liquid pollen germination medium) for 1 h. After fixation, pollen tubes were washed three times in wash buffer (50 mM Tris-HCl, pH7.4, 200 mM NaCl, 14% [w/v] sucrose, 0.05% [w/v] Nonidet P-40) for 10 min and reacted with 200 nM Alexa 488 phalloidin in wash buffer supplemented with 1% (w/v) bovine serum albumen (BSA) overnight in darkness at 4 °C. The stained pollen tubes were washed twice for 10 min with wash buffer and actin was observed using a confocal laser scanning microscope (Leica; TCS SP5) equipped with a ×100 oil objective (1.4-numerical aperture HC PLAN).

**Actin imaging and analysis**. The images of actin in pollen tubes were captured using a Leica TCS SP5 confocal laser scanning microscope with a ×100 oil objective (1.4 numerical aperture HC PLAN). Fluorescence of Alexa 488 phalloidin was excited with a 488-nm argon laser and the emission wavelength was set at 500–545 nm. Pollen tube images were generated by stacking all optical sections at a z-step size of 0.5 µm and analyzed using the LAS AF Lite software (Leica; Version 2.6.0). To measure the distance from the tip of the pollen tube to the start of the dense sub-apical actin fringe, five individual lines were drawn parallel to the pollen tube, from the plasma membrane at the tip to the end of an actin filament structure. The length

is expressed as the average value calculated for the distance from the plasma membrane to the point of maximum fluorescence intensity on each of the five lines.

**Measurement of plasma membrane potential.** For Annine-6-plus membrane potential measurements, wild-type and mutant pollen grains were collected and germinated as described above. Pollen grains were incubated at 21.5 °C for at least 2 h before adding the membrane potential dye Annine-6-plus to a final concentration of 10 μg.ml$^{-1}$. Fluorescence images were acquired on a Zeiss LSM 980 Airyscan 2 Laser Scanning Confocal using a x63 NA1.4 oil immersion objective, in the Super-Resolution mode, and were processed with Airy Scan Super Resolution processing software in Zen lite. The estimated resolution was better than 0.2 μm. Filter settings were as follows: excitation, 475 nm and emission, 499–549 nm (dual-band BP) and 573–627 nm (BP). Images were analyzed in ImageJ by drawing transects that were 2 μm apart and perpendicular to the pollen tube growth axis (see Supplementary Fig. 12). Membrane fluorescence values were averaged from the three maxima pixels in each transect for both the left and right sides of the shank (Supplementary Fig. 12c), corresponding to better than a tenth of the theoretical resolution, and well above the Nyquist criteria in terms of resolution and device pixel resolution oversampling. Images were acquired in strictly the same settings and normalized between different biological replicates by (1) background fluorescence and (2) average cytosolic fluorescence. Final values are the median of fluorescent intensities between 10–20 μm and 40–50 μm from the tip (D, E).

For analysis using DiBAC$_4$(3), pollen tubes of wild-type and aha6/8/9 plants were germinated and grown for 3 h and thereafter stained with 2.5 μM DiBAC$_4$(3) for 10 min right before observation. Controls for endocytosis of the dye were made by treating the pollen tubes with 0.8 μM wortmannin in the liquid pollen germination medium for 1 h after pollen growth and staining with FM4-64 (2 μM) for 10 min immediately before observation. FM4-64 and DiBAC$_4$(3) were excited with a 514-nm and 488-nm argon laser, respectively. The emission wavelength of FM4-64 was set at 620–660 nm, and that of DiBAC$_4$(3) at 515–565 nm using a confocal microscope (Leica SP5). DiBAC$_4$(3) fluorescence was measured as the intensity in the cytosol area in the pollen tube tip and shank (40–50 μm from the tip) using ImageJ software (https://imagej.nih.gov/ij/).

**Accession numbers.** Sequence data from this article can be found in GenBank/EMBL or the Arabidopsis Genome Initiative database under the following accession numbers: AHA6 (AT2G07560), AHA7 (AT3G60330), AHA8 (AT3G42640), and AHA9 (AT1G80660). The following T-DNA insertion lines were used in this study: SAIL_1293_D09 (aha6-1), SALK_042485 (aha7-1), SALK_105742C (aha7-2), SAIL_823_E09 (aha8-1), SALK_208753 (aha8-3), SALK_064821 (aha9-1; showed aborted male and female gametophytes is heterozygous lines; probably having chromosomal rearrangement and was not further analyzed), SALK_127945 (aha9-4), and GK-764H12 (aha9-5).

**Reporting summary.** Further information on research design is available in the Nature Research Reporting Summary linked to this article.

## Data availability
All data supporting the findings of this study are included in this published article, its Supplementary Information, and the Source Data file. Any other data and material will be made available upon reasonable request.

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

## Acknowledgements

M.P. laboratory was supported by grants from the Danish National Research Centre "Centre for Membrane Pumps in Cells and Disease" (PumpKin) and Innovation Fund Denmark (LESSISMORE). J.A.F. laboratory was supported in PT by the Fundação para a Ciência e Tecnologia (PTDC/BIA-PLA/4018/2012) and in the US by the National Science Foundation (MCB 1616437/2016 and 1930165/2019) and the University of Maryland. D.D. was funded by the grant 19/23343-7 from the São Paulo Research Foundation (FAPESP). Purchase of the Zeiss LSM 980 Airyscan 2 was supported by the National Institutes of Health award 1S10OD025223-01A1 to the Imaging Core of the Univ. Maryland.

## Author contributions

R.H., M.T.P., L.O., D.D., M.H., J.F., and M.P. designed the research; R.H., M.T.P., L.O., D.D., M.H., C.N., J.P., P.L., and C.C. performed the research. R.H., M.T.P., L.O., D.D., M.H., C.N., J.P., P.L., J.F., and M.P. analyzed the data. R.H., M.T.P., L.O., D.D., M.H., J.F., and M.P. wrote the paper.

## Competing interests

The authors declare no competing interests.
