## [Peer Review File · Nature Communications]

Reviewers' comments:

Reviewer #1 (Remarks to the Author):

Hoffmann et al. reports genetic and phenotypic characterization of loss of function *aha* mutants in pollen tube growth in *Arabidopsis*. The authors present pollen tube growth defect of *aha6/8/9* mutants, which correlates with lower cytoplasmic pH values, cytoplasmic pH oscillation profiles along the tube, actin organization, and pollen tube shapes. No single mutants of *aha6*, *7*, *8*, or *9* showed phenotypic alterations compared with wildtype plants. On the other hands, they demonstrate that triple knockout mutations of *aha6/8/9* dramatically reduced pollen tube growth thus seed production. Plasma membrane H⁺-ATPase is an essential ion pump in yeast and plants. In the *Arabidopsis* genome, *AHA* forms a gene family whereby cell-type specificity as well as overall protein expression levels determine the function of each isoforms. In this study, the authors examine pollen expressed *AHA* isoforms.

In molecular genetic characterization, their results showed that *aha6/8/9* triple homozygous mutant can be viable. Description and characterization of mutant alleles are confusing. In the diagram in Fig 2a, *aha8-1* has an insertion at one of the exon with which one would expect this allele is null. In Fig. 2b and c, *aha8-1* plant expresses *AHA8* mRNA and *aha8-3* plant appeared to be a null mutant. But they have chosen to work *aha8-1* which is a weak allele based on their characterization. It would be useful if they can provide more explanation for characterization of *aha8* mutant alleles. They mention quadruple *aha6/7/8/9* mutant does not exist. This is an ambiguous statement because they did not show numbers of plants screened nor statistics to support this. Ideally they should have shown molecular complementation of triple or quadruple mutants. Also GFP tagging of *AHA6*, *8*, and *9*, shown in Fig. 1a, are they triple mutant background? It would make sense if they show *AHAX*-GFP expression in the mutant background and functional complementation. This was not clearly mentioned in the text. Another possibility is using yeast heterogenous expression system to test the function of the tagged *AHAXs*. In Fig. 1b and in the text, they describe that the full length *AHA6*, *8*, and *9* do not complement *pma1* knockout yeast strain, because of the presence of the C-terminal auto-inhibitory domain. Is this statement true only for *AHA6*, *8*, and *9*? Or also applicable to other *AHAs*? *AHA2* or tobacco H⁺-ATPase gene expression were reported by the same author and successful complementation was shown using *pma1* complementation assay. Rudashevskaya et al. 2012 showed *AHA2* full length can complement *pma1*. Please comment on relationship of functional complementation assay in yeast and plants.

Fig.2 f and h. The results shown in seed counts (f) and growth rate (h) do not seem to agree. In seed counts, *aha6/8* showed the striking defective phenotype, but it does not show defect in the growth rate assay. On the other hand, *aha6/9* mutant shows the moderate defect in seed counts, but reduced growth rate compared with wildtype. Please double check the genotype comparison. If this is the case, there seems to be specific roles for *AHA8* and *AHA9* during pollen tube elongation and fertilization, which needs to be discussed.

For in vitro pollen tube assay, they have used the synthetic media at pH7.5. Does this assay reflect what pollen tubes experience in the style in vivo? The original lily pollen paper used pH 5.7. It might be useful to mention the evolution of pollen tube assay system, although the space might be limited. Can they do their pHluorin assay in the *Arabidopsis* stigma surface to check pH of the apoplast during pollen germination in vivo?

It would provide a stronger support if they provide data on imaging/measurement of extracellular pH, using either pHluorin or pH sensitive reporter dye. As expected, *aha* mutant shows acidic cytoplasmic pH in this report, though cytoplasmic pH can be affected by other transporter activity located at endomembranes, which could be difficult to dissect but will be worth to bring readers attention. They may also comment that the lack of *AHA*-GFP signal at the tip. GFP fluorescent emission is influenced by pH and other ions such as chloride. Have they tested immuno stain to

check AHA protein level in addition to imaging methods?

Three other points severely detract from this reviewers' enthusiasm for the rigor of the work and for the rigor of the mechanistic interpretation.

First, The pump has four products: cytoplasmic pH, cell wall pH (these two form the pH gradient but this gradient can be derived from different sets of inside and outside pH values), the pH gradient and the membrane potential. As discussed above, although the authors report measurements of pH inside no where do i see a report on the membrane potential or the external pH.. Thus any interpretation of the role of anion fluxes or cation fluxes or any fluxes without knowing how what the changes in the membrane potential and delta pH are as a function of genotype and position in the tube and time after germination of tube, are not possible and equivocal.

Second, the measurements of turgor pressure are not really turgor pressure. They are measuring the osmotic potential. One needs to directly measure pressure with a pressure probe and then with the osmotic potential of the cell contents known, the total water potential can be determined. But incipient plasmolysis does not measure turgor pressure. It measures the osmotic potential outside the cell at which the cell loses turgor but this outside osmotic potential is not turgor pressure. A pressure probe must be used to make any conclusions on turgor pressure.

Third, and this one may seem trivial but is a bit disturbing since it is the first time this reviewer has seen it. The authors refer to AHA's as Autoinhibited H-Atases..but the term was first coined, and has been used consistently since then, to refer to Arabidopsis H⁺-ATPases...that is why one must use NtAHA1 etc to denote the tobacco orthologues, for example (Nt is Nicotiana tabacum). While it may be true that many or all of the AHAs have a c terminal domain that acts as an autoinibitory domain under some conditions (eg expression in yeast and complementation of yeast pma1 knockouts), it is not at all clear to this reviewer that this should be the defining characteristic of this set of pumps. What is especially annoying is that this comes out of nowhere and for folks following the literature, brings in considerable confusion without a clear explanation for why it is put forward.

In conclusion, while the relationship between tube growth and genotype of the pump seems to be straightforward the interpretation of this, especially in relation to turgor pressure and anion fluxes, without a clear measurement of the turgor pressure at various points in the tube, and without measurements of membrane potential, the interpretation and model simply cannot stand. The pump's role at the plasma membrane is fundamental for all of the other cotransporters and channels that one must be VERY careful about interpreting a phenotype without a precise measurement of the properties such as membrane potential and turgor pressure. It is not surprising to see cytoskeletal properties altered in the mutant but mechanistically ruling our turgor pressure and incorporating anion fluxes specifically wihtout measurements of the real pressure and the membrane potential seem to set the rigor of this paper too low the bar of what one expects in a paper in Nature Communications.

Other points:

- Supplemental figure 1, for mRNA expression profiles and comparisons among AHA6, 7, 8, and 9, it will be useful to show the expression levels of the four AHAs in pollen. The current figures shows the expression of each genes in various developmental stages, which supports they are pollen specific isoforms, but from this figure one cannot directly see a comparison of pollen expression among four genes (i.e., which gene has the highest expression in pollen?)

- Fig.1. The image for AHA9-GFP is truncated at the tip of the pollen tube. It would be nice to be able to see the entire pollen tube.

- Page 7, line 16. Overexpression of AHA7 partially rescued pollen tube defect of *aha6/8/9*. No data shown to support this statement. Also no information provided as to the promoter or plasmid construct

Reviewer #2 (Remarks to the Author):

The Hoffman et al. study furthers the demonstration of H⁺-ATPases as key regulator of growth, and in this case, the study focuses on polarize growth in pollen tubes, where there is a rich history of ionic regulation, in particular by H⁺ and Ca²⁺. The study was carried by long standing expert groups in H⁺-ATPases and regulation of ion homeostasis in pollen tubes.

The premise of the study is straight-forward, based on the characterization of double and triple *aha* mutants in Arabidopsis, *ara6* being at the core, its combination with one or two of two additional mutant *ahas* (8, 9, and to a lesser extent 7). Not surprisingly pollen tube growth became increasingly compromised to the extent of reaching close to male sterile (Fig. 1,2) . That's where the physiological aspects of characterizing the H⁺-related parameters began. While the electrophysiology studies were rather extensive, the results were demonstrably presented in several summary plots (Fig 3,4). The data trends are generally in agreement with the order of combined mutations and the severity of tube growth properties (while not precisely linear, the data are reasonably cohesive). As a reviewer far from being an expert in these physiological approaches, the summary data presented here is adequate to relay the authors conclusion from these data. I'll leave for the experts to decipher the actual data acquired and the statistically analyses. Of Fig. 3,4, I wish Fig. 4C to be more accessible. Supplemental Fig. 9 also.

Fig. 5 shows disruption in actin accompanies loss of *aha*-induced pollen tube growth defects. While I understand the wish to connect disrupt H⁺ status to the key structure that support pollen tube growth, the demonstration is hardly adequate to rely a direct causal relationship, although it's hard to recommend another target to examine.

Overall I find it to be a huge amount of meticulously collected data, both on the genetics and reproduction characterization and the pollen tube growth and physiology analyses. The findings are consistent with what one would expect with AHA deficiency.

I suggest some efforts in revising sentence structure might help the reading of the manuscript. Examples:

Pg. 2, Lines 11-14: perhaps try :“ Tubes lacking AHAt displayed reduced cytosolic pH, tip-to-shank protein gradients..... ” the phrase “which were synchrhoized with ossiclations in growth rate”.

Pg. 4, Lines 9-11: either rearrange the phrases of separate into two sentences.

Reviewer #3 (Remarks to the Author):

Involvement of H⁺ -ATPases in pollen tube growth has long been suggested. However, direct evidence linking AHAs with pollen tube growth was lacking due to its high degree of genetic redundancy. In this MS, authors are providing solid genetic evidence that PM H⁺ -ATPases play a pivotal role in pollen tube growth. I congratulate authors for taking multiple experimental approaches to demonstrate the functional role of AHA6/8/9 in modulating H⁺ fluxes and intracellular pH dynamics. The findings of this study will be highly useful to refine tip-growing theories. Overall this MS is well written and all the conclusions are well supported with actual data. Though the involvement of AHAs in regulating H⁺ fluxes is demonstrated in this MS, it would be nice to show how these H⁺ fluxes alter the membrane potential in tip and shank. Based on the H⁺

flux data the tip should show depolarisation and the shank should show hyperpolarisation. The AHA mutants' also should have depolarised membrane potential compare to wild type in shank.

Some other minor points to consider

P 5 In 3, 6 and 7 the figure you are mentioning here is 1a NOT 1b.

P 7 In 15-16 Indeed, overexpression of AHA7 partially rescued the pollen tube growth defect of *aha6/8/9*.- No data has been presented to support this claim.

P21 In 30-34. It is critical to provide ion-selective cocktail catalogue number used for making H⁺ and Cl⁻ selective electrodes.

P35 Figure 1 legend. Please modify based on the actual figure. Panel a and b are swapped.

P 36 Figure 3 legend. On panel C please mention negative sign means influx and the positive sign means efflux.

Please revise supplementary figure 9 legend panel information.

Jayakumar Bose

NCOMMS-19-90098A

Reviewer #1

Comment by reviewer:

In molecular genetic characterization, their results showed that *aha6/8/9* triple homozygous mutant can be viable. Description and characterization of mutant alleles are confusing. In the diagram in Fig 2a, *aha8-1* has an insertion at one of the exon with which one would expect this allele is null. In Fig. 2b and c, *aha8-1* plant expresses *AHA8* mRNA and *aha8-3* plant appeared to be a null mutant. But they have chosen to work *aha8-1* which is a weak allele based on their characterization. It would be useful if they can provide more explanation for characterization of *aha8* mutant alleles.

Reply:

The reviewer states correctly that *aha8-1* has an insertion in an exon. The insertion is in the middle of the coding sequence and we verified the absence of full-length cDNA (Figure 2c). Additional verification of the effective knockout of the *AHA8* allele in *aha8-1* is provided by the very early pollen tube growth arrest and the resulting reduction in the number of seeds per silique of the *aha6/aha8* double mutant (Figure 2e and f), which is fully rescued by inserting back *AHA8* under its native promoter (Supplementary Figure 4).

Comment by reviewer:

They mention quadruple *aha6/7/8/9* mutant does not exist. This is an ambiguous statement because they did not show numbers of plants screened nor statistics to support this. Ideally they should have shown molecular complementation of triple or quadruple mutants.

Reply:

We wrote in our manuscript that “quadruple mutants were not identifiable” and provide in Supplementary Fig. 7c genotyping data from over 500 triple mutant plants that were pollinated with pollen from plants heterozygous for the fourth allele. We show successful complementation of *aha6*, *aha8* and *aha9* in Supplementary Fig. 4.

Comment by reviewer:

Also GFP tagging of *AHA6*, 8, and 9, shown in Fig. 1a, are they triple mutant background? It would make sense if they show *AHAx-GFP* expression in the mutant background and functional complementation. This was not clearly mentioned in the text.

Reply:

The pollen tubes showing GFP expression in Fig. 1 are double mutants that were complemented with one respective *promoter::gene::GFP* construct. In the *aha6/8* double mutant, *AHA6::GFP* fully complemented the loss of *AHA6* (Supplementary Figure 4a). *AHA8::GFP* partially rescued the mutant phenotype (whereas *AHA8* without GFP fully rescued the mutant phenotype). *AHA9::GFP* fully rescued the *aha6/9* phenotype of reduced seed setting (Supplementary Figure 4a). We acknowledge that this was not clear in the

previous version of our manuscript and have made changes in the revised manuscript that allow the reader to better follow our experimental procedures.

Comment by reviewer:

In Fig. 1b and in the text, they describe that the full length AHA6, 8, and 9 do not complement *pma1* knockout yeast strain, because of the presence of the C-terminal auto-inhibitory domain. Is this statement true only for AHA6, 8, and 9? Or also applicable to other AHAs? AHA2 or tobacco H⁺-ATPase gene expression were reported by the same author and successful complementation was shown using *pma* complementation assay. Rudashevskaya *et al.* 2012 showed AHA2 full length can complement *pma1*. Please comment on relationship of functional complementation assay in yeast and plants.

Reply:

The statement is true for other AHAs and this has now been made clear in the text. The autoinhibitory C-terminal domain of AHAs does not block pump activity, but controls the degree of coupling between ATP hydrolysis and H⁺ pumping, i.e. pumping efficiency (Pedersen *et al.*, 2015, J Biol Chem. 290, 20396-406). Thus, the ability of full-length AHAs to complement *S. cerevisiae pma1* depends on the size of the electrochemical H⁺ gradient that the pumps are pumping against and the duration of the growth assay. When transformed yeast cells are grown on standard synthetic growth medium (pH 5.5), those expressing C-terminally truncated AHA2 grow well—almost as well as cells expressing PMA1. Cells expressing full-length AHA2 also grow, albeit much more slowly than those expressing the full-length pump (Fig. 1C: Palmgren & Christensen, 1993, FEBS Lett. 317, 216-222; Fig. 1: Axelsen *et al.*, 1999, Biochemistry 38, 7227-7234). However, with time, a culture of AHA2-expressing yeast cells will eventually reach the same density as those expressing PMA1. In Rudashevskaya *et al.* (2012), no comparison with PMA1 was made. Such a comparison is shown in Fig. 1 in Axelsen *et al.* (1999), where it can be seen that even though full-length AHA2 can complement *pma1* to some degree at an external pH of 5.5, growth is challenged at more acidic external pH. Cells expressing C-terminally truncated AHA2 grow much better than cells expressing full-length AHA2 at an external pH of 4.1 or 3.6. At an external pH of 3.1, even the truncated AHA2 can barely complement a yeast strain expressing PMA1.

Comment by reviewer:

Fig.2 f and h. The results shown in seed counts (f) and growth rate (h) do not seem to agree. In seed counts, *aha6/8* showed the striking defective phenotype, but it does not show defect in the growth rate assay. On the other hand, *aha6/9* mutant shows the moderate defect in seed counts, but reduced growth rate compared with wildtype. Please double check the genotype comparison. If this is the case, there seems to be specific roles for AHA8 and AHA9 during pollen tube elongation and fertilization, which needs to be discussed.

Reply:

We purposefully refrained from discussing these results in detail, to keep the focus of the general conclusions on the function of plasma membrane proton pumps in pollen tubes. However, we acknowledge that these results are noteworthy and have therefore added the following text in the revised manuscript:

“When self-fertilized, the three combinations of double mutants (*aha6/8*, *aha6/9*, and *aha8/9*) showed different degrees of early pollen tube growth arrest (Fig. 2c, d), reduced pollen tube elongation rates (Fig. 2d), and reduced fertility (Fig. 2e). *aha6/8* pollen tubes grew at a rate similar to that of the wild type, but terminated growth much earlier, whereas tubes of *aha6/9* germinated later and grew slower than wild-type tubes but grew almost as long as those of the wild type. This suggests that AHA6 and AHA9 together function in pollen tube germination and the early phases of tube elongation, whereas AHA6 and AHA8 together are important for sustained pollen tube growth.”

Comment by reviewer:

For in vitro pollen tube assay, they have used the synthetic media at pH7.5. Does this assay reflect what pollen tubes experience in the style in vivo? The original lily pollen paper used pH 5.7. It might be useful to mention the evolution of pollen tube assay system, although the space might be limited. Can they do their pHluorin assay in the Arabidopsis stigma surface to check pH of the apoplast during pollen germination in vivo?

Reply:

The reviewer is correct that acidic conditions are optimal for the germination of pollen from lily, tobacco, petunia, tomato, and most flowering plants, and that the pollen from some species does not germinate at all at neutral pH. Arabidopsis, however, is unusual, to the extent that it germinates very poorly on artificial medium (accession variable, with Landsberg having the lowest pollen germination rate, followed by Columbia and then C24), and for decades researchers have struggled to develop reliable germination protocols for Arabidopsis pollen, because most groups have mistakenly assumed that an acidic medium, such as Brewbaker and Kwack’s classic medium, would be suitable. At-Columbia pollen was only reliably and reproducibly germinated when either (1) the pollen was germinated at high density in the presence of stylar parts or (2) once media with neutral pHs were used. The need for neutral pH is not completely unheard of (e.g., for *Papaver* pollen) and further systematic investigation defined temperature as another critical parameter. The present golden standard was described by Boavida and McCormick (2007, *Plant J.*) with minor variations. In the optimized protocol we developed for this report, pollen is collected without dissection of the anthers, by centrifugation of whole flowers in germination medium at pH 7.5–7.6, and re-suspended at high density. Under these conditions, germination is reliably over 80%, and supports robust statistical analysis. In the context of the present paper, these conditions actually suggest two major conclusions that we would like to point out. First, the common concept that “pollen tubes have cytosolic acid pH in the tip” should henceforth be re-phrased to “cytosolic pH in the tip of pollen tubes EQUILIBRATES with the germination medium, and the cytosolic pH in the shank is determined by the action of AHAs”. Second, given the universal dependency of all genetic probes (e.g., pHluorin) on pH, the cytosolic measurements we present in this manuscript are likely to be the most accurate ever described in pollen tubes.

Comment by reviewer:

It would provide a stronger support if they provide data on imaging/measurement of extracellular pH, using either pHluorin or pH sensitive reporter dye. As expected, *aha* mutant

shows acidic cytoplasmic pH in this report, though cytoplasmic pH can be affected by other transporter activity located at endomembranes, which could be difficult to dissect but will be worth to bring readers attention.

Reply:

We respectfully disagree with the reviewer on this point. We have indeed provided extracellular pH data, and of much better quality than would have been obtained using either of the imaging methods proposed. Our data were obtained using an H⁺-specific vibrating probe, which measures, with unsurpassable precision, the gradients in the vicinity of the cell wall. Therefore, when we refer to an influx at the tip, that means we detect an alkaline gradient in the 10- μ m region orthogonally adjacent to tip where we vibrate the [H⁺] probe. The magnitude of such a gradient is then used, after subtracting the background pH of the medium, to calculate the fluxes through Fick's laws of diffusion. In absolute terms, over 10 μ m, the gradients are usually below 0.01 pH units on a pH 7 background, which would make them undetectable using any of the imaging methods and explain why the many groups that have tried such imaging approaches have not succeeded. The vibrating probe, as documented, is sensitive enough to detect such small pH gradients, with an S/N ratio of more than two orders of magnitude, i.e., twice the Nyquist criteria. We do not believe that endomembranes account for the differences observed in our mutants.

Comment by reviewer:

They may also comment that the lack of AHA-GFP signal at the tip. GFP fluorescent emission is influenced by pH and other ions such as chloride. Have they tested immuno stain to check AHA protein level in addition to imaging methods?

Reply:

The GFP localization of AHA in tobacco pollen tubes (Cortal *et al.* 2008) was also identified by immunostaining by the group of Marc Boutry (ref#15 on the original manuscript and Lefebvre *et al.* 2006, Plant Cell), providing proof that AHA is absent in the growing tip. As elaborated above, the issue of pH is not at stake for Arabidopsis given that, contrary to other species that require acidic pH for germination, At pollen germinates at neutral pH, and, as we show in the present manuscript, has a cytosolic neutral pH. Chloride values at the tip are at the same level as in the more distal shank, with elevation in the more proximal shank, as we have described before for tobacco pollen (Guttermuth *et al.*, 2013, 2018), a pattern which does not fit, on the one hand, with our present results in the localization of the different AHAs and, on the other, with its exclusion from the tip. The results we present in this manuscript thus fully agree with those from Cortal *et al.*, i.e. there is no structural correlation between pH and Cl⁻ and the GFP localization of AHAs, and we therefore have no reason to believe that the absence of Arabidopsis GFP-tagged AHA from the growing tip is a result of the ionic environment.

Comment by reviewer:

Three other points severely detract from this reviewers' enthusiasm for the rigor of the work and for the rigor of the mechanistic interpretation.

First, the pump has four products: cytoplasmic pH, cell wall pH (these two form the pH gradient but this gradient can be derived from different sets of inside and outside pH values), the pH gradient and the membrane potential. As discussed above, although the authors report measurements of pH inside no where do i see a report on the membrane potential or the external pH. Thus any interpretation of the role of anion fluxes or cation fluxes or any fluxes without knowing how what the changes in the membrane potential and delta pH are as a function of genotype and position in the tube and time after germination of tube, are not possible and equivocal.

Reply:

Measurements of membrane potential:

We agree with the Reviewers #1 and #3 that including data on the effects of AHA loss on the membrane potential of pollen tubes would improve the quality of the work.

In the revised manuscript, we have added new data using ANNINE-6-plus for quantitative comparisons of the membrane potentials in wild-type and mutant pollen tubes based on fluorescence intensity. ANNINE-6-plus is a voltage-sensitive fast-response dye that has been used for sensitive optical recording of neuronal excitation. While methodological proof-of-principle demonstrations have been published (Flickinger *et al.*, 2010, *Protoplasma*; Berghoefer *et al.*, 2012, *Plant Sign.Behav.*), to the best of our knowledge, this is the first application of this dye in a functional study involving plant cell phenotyping. We would like to emphasize this point, since under optimized imaging conditions ANNINE dyes are highly sensitive (0.5%/mV), have negligible bleaching and phototoxicity, a linear response to membrane potentials of over 120 mV, and a temporal resolution that is faster than the optical imaging devices currently used in biology (in the order of nanoseconds). Furthermore, ANNINE-6-plus has been reported to stably allow voltage imaging in neurons for over 2 weeks (Kuhn and Roome 2019, *Front. Cell. Neurosci.*).

In addition, we took advantage of the near-super-resolution properties of the latest Airy-Disk Zeiss technology. This technology allowed full closure of the confocal pinhole, pushing the x-y planar resolution to $< 0.2 \mu\text{m}$, and the z-optical thickness to $< 0.4 \mu\text{m}$, resulting in an optimally resolved plasma membrane signal that allows for the quantification of fluorescence intensity based on $< 0.05\text{-}\mu\text{m}$ pixels in the middle of the plasma membrane plane, i.e., better than the Nyquist criteria for the calculated pixel number used in our acquisition protocol.

We subsequently developed a robust quantification protocol, including normalization of biological replicates by ratioing the plasma membrane signal with the background noise and cytosol autofluorescence levels. While we cannot offer a robust protocol of absolute calibration, which will likely take a substantial period of time to optimize to the level of full confidence, we are very confident that the absolute levels of fluorescence quantified are comparable and biologically meaningful. Furthermore, we focused strictly on the sub-apical zone of the pollen tube (10–20 μm distal from the tip, where AHA labeling fades away) and the AHA-stabilized shank (40–50 μm from the tip). We deliberately avoided quantifications in the tip, given its extremely fast rates of lipid phase recycling, the impact of which, so far,

has not been addressed in terms of clearing the probe concentration in the plane of the membrane, thus creating false polarization measurements by depletion of the probes.

From this analysis, it is now clear that all the mutants have less hyperpolarized membrane potentials than the wild type, with *aha6/9* and *aha6/8/9* having much lower (<50% of the wt) fluorescence.

To obtain independent evidence for this finding, we further used the slow-response oxonol dye DiBAC₄(3) to measure membrane potentials in growing pollen tubes. DiBAC₄(3) is a voltage-sensitive dye that enters depolarized cells and is excluded from hyperpolarized cells. As pollen tubes have high rates of endocytosis at the tip, endocytosis contributes to the uptake of slow probe, which previous work using this dye for measurements of membrane potentials in pollen tubes has not accounted for. However, when wortmannin was included in the assay to block endocytosis, we observed that the *aha6/8/9* triple mutant had a much higher degree of DiBAC₄(3) uptake compared to the wild type, which is indicative of a less negative potential. The effect of Wortmannin on endocytosis was validated by inhibition of FM4-64 uptake, as previously demonstrated (Zhang *et al.*, 2010, Plant Physiol.).

We believe that our demonstration that the fast probe ANNINE-6plus can be used to measure changes in pollen tube membrane potentials is a long-sought and important contribution to the field. In the absence of a robust calibration one must be careful, however, from translating fluorescence changes of the dyes into actual membrane potentials as done by some researchers in the field. Pollen tubes have oscillatory phenomena that induce large variations in ion concentration, similar to those observed in excitable cells such as neurons, and are incredibly dynamic at the level of membrane recycling and cytoskeletal organization at the tip. Therefore, we would not feel comfortable quantifying the membrane potentials until a reliable calibration protocol has been established, which is likely to involve multiple probes and a combination of electrophysiology and imaging that we are not yet close to achieving.

However, given the reported plasma membrane potential of -127 mV (with short interspersed depolarizations of 10–20 mV, Mouline *et al.*, 2002), a loss of 50% fluorescence in the mutant would mean that the membrane potential could dip to less than -70 mV. We base this extrapolation on the outstanding linearity of the probe at least up to -80 mV, but we note that calibrations are lacking for the hyperpolarization levels of pollen tubes, which, if anything, would only affect the inferred value by default, i.e., if the dye becomes non-linear over -100 mV, then half of the fluorescence would be LESS than half of the membrane potential. Based in this line of reasoning, we allude to these values in the main manuscript as indicative landmarks.

In conclusion, we believe that the quantitative accuracy of ANNINE-6-plus and the confirmation of our findings with DiBAC₄(3) provide unequivocal evidence that AHA mutants, which are impaired in growth and fertility, have much less negative membrane potentials, thus substantiating a mechanistic role for AHA in pollen tube growth through plasma membrane hyperpolarization energization.

External pH

As explained above, we provide precise and detailed measurements of the extracellular pH, which we transform into fluxes for ease of interpretation. Therefore, the only parameter missing from the reviewer's list is the cell wall pH. We argue that this parameter is probably not that relevant, and, to the best of our knowledge, should grossly equilibrate with the medium.

We base this argument on the following points:

(1) At least three groups with reputable credentials in imaging have tried to image pH gradients around and within cell walls of WT pollen tubes without much success in at least three different species (Arabidopsis, tobacco, and lily).

(2) The stereotypical textbook model of cell wall organization in plants, i.e., highly ordered and rich in negative charge, with networks of cellulose fibrils, does not apply to pollen tubes. Rather, as revealed by fast-freezing scanning electron microscopy research conducted by Derksen *et al.* (2011, Plant J.), the pollen tube cell wall is composed of a "...lattice consisting of longer fibers, approximately 10–15 nm wide, with rod-like, thinner interconnections at angles of approximately 90° with the longer fibers. Such architecture may reflect functional needs with respect to porosity and mechanical strength. The wall does not form a mechanical barrier to interaction with the environment and is gained at low cost." Hence, one could describe the pollen tube cell wall as a highly porous, sponge-like structure, rather than the cellulosic structure of the typical diffuse growing vegetative cell wall described in textbooks. This makes sense, if one considers the need of pollen tubes to incorporate proteins of various size encountered along the pistil path, at a diffusion rate that is fast enough to facilitate binding to receptors and trigger the signaling pathways that bring about critical changes in growth direction as the pollen tube rapidly grows towards the ovule. Such a porous cell wall is unlikely to sustain an internal pH gradient; it seems much more likely that the pollen tube cell wall equilibrates with the immediate vicinity, the pH of which we measure with the vibrating probe.

Comment by reviewer:

Second, the measurements of turgor pressure are not really turgor pressure. They are measuring the osmotic potential. One needs to directly measure pressure with a pressure probe and then with the osmotic potential of the cell contents known, the total water potential can be determined. But incipient plasmolysis does not measure turgor pressure. It measures the osmotic potential outside the cell at which the cell loses turgor but this outside osmotic potential is not turgor pressure. A pressure probe must be used to make any conclusions on turgor pressure.

Reply:

Measurements of turgor pressure:

We have now made it clear in the text that we have used incipient plasmolysis as an indirect proxy for turgor as opposed to a direct measure, and have framed the conclusions in light of any assumptions that underpin this approach.

A major problem relates to DIRECTLY measuring turgor pressure in Arabidopsis pollen tubes. Reviewer 1 had this comment regarding measuring turgor pressure: "Second, the measurements of turgor pressure are not really turgor pressure. They are measuring the osmotic potential. One needs to directly measure pressure with a pressure probe and then with the osmotic potential of the cell contents known, the total water potential can be determined. But incipient plasmolysis does not measure turgor pressure. It measures the osmotic potential outside the cell at which the cell loses turgor but this outside osmotic potential is not turgor pressure. A pressure probe MUST be used to make any conclusions on turgor pressure."(capitals added by us).

We unfortunately do not feel that this is a feasible option. To date, only one publication, from Gerhard Obermeyer's lab (Benkert *et al.*, 1997), has reported the use of a pressure probe to measure pollen tube turgor. However, this analysis was performed in *Lilium longiflorum*, which was traditionally used as a cell biology model organism due to its wide (diameter ca. 15 μm), sturdy, and straight pollen tubes. This is in striking contrast to Arabidopsis pollen tubes, which are much thinner (diameter ca. 5 μm) and highly sensitive to any manipulation, especially in the Columbia ecotype. Reliable results require the pressure probe to have a tip diameter of ca. 2 μm (Benkert *et al.*, 1997), which makes it virtually impossible to use this method in Arabidopsis pollen tubes. In fact, one of us (JF) visited Obermeyer's lab in 2001 and attempted to perform these experiments in tobacco pollen tubes, without success. Even though tobacco pollen tubes are 50% thicker (ca. ~ 8 μm diameter) and much more robust than those from Arabidopsis, most of the pollen tubes stopped growing or burst when impaled with the oil-filled ~ 2 μm -thick pipettes, and we obtained no reliable measurements. Based on this previous experience, we respectfully think that Reviewer 1 may not fully appreciate the limitations of working with a pressure probe in Arabidopsis pollen tubes.

This problem has also been described in a previous Nature Communications paper and, as we have done here, these authors used incipient plasmolysis as a proxy (<https://www.nature.com/articles/ncomms7030>). Other INDIRECT methods to measure turgor pressure have been published, but the only reliable advance came from the laboratory of Ueli Grossniklaus, which developed a novel non-invasive indentation method. However, so far, this method has only been applied to *Lilium longiflorum* pollen tubes (Vogler *et al.* 2013, Plant J., 73:617, and Burri *et al.*, 2019 *et al.* <https://ieeexplore.ieee.org/abstract/document/8610003>), and firm conclusions depend on the determination of the cell wall thickness, which would require another parallel project. To the best of our knowledge, no other method has been reported to measure Arabidopsis pollen turgor.

There is a consensus in the literature that turgor pressure can be deduced from osmotic potential using indirect methods, as we have done here (reviewed in

<https://www.ncbi.nlm.nih.gov/pmc/articles/PMC4204789/>) and, even if this approach raises problems in terms of absolute values, it can be used to identify differences in phenotypes in Arabidopsis pollen tubes, as described in the previously cited Nature Communications publication.

Thus, we have refrained from using the pressure probe, and report turgor on the basis of indirect methods. We now clearly stated in the text that the method is indirect.

As a final note, we would like also to address the rationale behind the reviewer's note. The relative values of turgor pressure and cell wall elasticity in lily pollen tubes were estimated by Vogler *et al.* (2012, Plant J.) to be around 0.3 MPa and 20–90 (depending on cell wall thickness) Mpa, respectively. Thus, it is not easy for us to get around the idea that two orders of magnitude difference in pressure between turgor and cell wall elasticity can be dramatically impacted by the turgor alterations that may result from differences in H⁺ pumping, even if affecting other osmotically active drivers like Cl⁻ or K⁺. Indeed, Benkert *et al.* (1997) determined that turgor in a population of pollen tubes may vary by more than an order of magnitude, without any change in growth rate. Even when manipulating the internal turgor with the pressure probe, pollen tube growth only slowed when turgor was close to zero. It follows that the simplest explanation of these results is that a minimal turgor is needed to sustain growth, but turgor is not the main propeller of pollen tube growth, only affecting it below a minimal threshold needed to sustain growth. Together with the demonstration we now add of less negative plasma membrane potentials in the mutant lines, we believe that the electrochemistry associated with this phenomenon is a much more reasonable mechanistic explanation for the phenotypes observed, and stand by our conviction that the absence of variations of turgor revealed by the incipient plasmolysis method is indeed a reasonable and somehow expected result.

Comment by reviewer:

Third, and this one may seem trivial but is a bit disturbing since it is the first time this reviewer has seen it. The authors refer to AHA's as Autoinhibited H-ATPases..but the term was first coined, and has been used consistently since then, to refer to Arabidopsis H⁺-ATPases...that is why one must use NtAHA1 etc to denote the tobacco orthologues, for example (Nt is Nicotiana tabacum). While it may be true that many or all of the AHAs have a c terminal domain that acts as an autoinhibitory domain under some conditions (eg expression in yeast and complementation of yeast pma1 knockouts), it is not at all clear to this reviewer that this should be the defining characteristic of this set of pumps. What is especially annoying is that this comes out of nowhere and for folks following the literature, brings in considerable confusion without a clear explanation for why it is put forward.

Reply:

It is true that when the first plasma membrane H⁺-ATPase gene was cloned, AHA was used to designate "Arabidopsis H⁺-ATPase" (Harper *et al.* PNAS 1989), but in 2003 the meaning of the abbreviation was changed to signify "Autoinhibited H⁺-ATPases" (<http://www.plantphysiol.org/content/132/2/618.short>). This change was made so that the same nomenclature system could be used for plasma membrane H⁺-ATPases from all plant

species. It should be noted that Jeff Harper, who cloned the first AHA, is an author of both of these papers.

Comment by reviewer:

Supplemental figure 1, for mRNA expression profiles and comparisons among AHA6, 7, 8, and 9, it will be useful to show the expression levels of the four AHAs in pollen. The current figures shows the expression of each genes in various developmental stages, which supports they are pollen specific isoforms, but from this figure one cannot directly see a comparison of pollen expression among four genes (i.e., which gene has the highest expression in pollen?)

Reply:

We thank the reviewer for pointing this out. Actually, the absolute expression values were shown in the original figure. After reviewing the figure, we understand that the information was printed too small to be easily legible. We have redesigned the figure to better highlight absolute expression values.

Comment by reviewer:

Fig.1. The image for AHA9-GFP is truncated at the tip of the pollen tube. It would be nice to be able to see the entire pollen tube.

Reply:

Thank you for pointing this out. The image has now been replaced with one in which the tip is clearly visible.

Comment by reviewer:

Page 7, line 16. Overexpression of AHA7 partially rescued pollen tube defect of *aha6/8/9*. No data shown to support this statement. Also no information provided as to the promoter or plasmid construct

Reply:

We have changed the sentence to improve clarity: "*aha6/7/8/9* quadruple mutants were not identifiable (Supplementary Fig. 7). This suggests that AHA7 functions redundantly with the other isoforms, and may weakly sustain pollen tube growth in the *aha6/8/9* mutant."

Reviewer #2

Comment by reviewer:

Overall I find it to be a huge amount of meticulously collected data, both on the genetics and reproduction characterization and the pollen tube growth and physiology analyses. The findings are consistent with what one would expect with AHA deficiency.

Reply:

Thank you.

Comment by reviewer:

Of Fig. 3,4, I wish Fig. 4C to be more accessible. Supplemental Fig. 9 also.

Reply:

We apologize, but do not understand this request. We have redesigned the figures somewhat and hope that this has improved their accessibility.

Comment by reviewer:

I suggest some efforts in revising sentence structure might help the reading of the manuscript. Examples:

Pg. 2, Lines 11-14: perhaps try :” Tubes lacking AHAt displayed reduced cytosolic pH, tip-to-shank protein gradients..... “ the phrase “which were synchrhoized with osscillations in growth rate”.

Pg. 4, Lines 9-11: either rearrange the phrases of separate into two sentences.

Reply:

Thank you. The language has now been revised.

Reviewer #3

Comment by reviewer:

Involvement of H⁺ -ATPases in pollen tube growth has long been suggested. However, direct evidence linking AHAs with pollen tube growth was lacking due to its high degree of genetic redundancy. In this MS, authors are providing solid genetic evidence that PM H⁺ -ATPases play a pivotal role in pollen tube growth. I congratulate authors for taking multiple experimental approaches to demonstrate the functional role of AHA6/8/9 in modulating H⁺ fluxes and intracellular pH dynamics. The findings of this study will be highly useful to refine tip-growing theories. Overall this MS is well written and all the conclusions are well supported with actual data.

Reply:

Thank you.

Comment by reviewer:

Though the involvement of AHAs in regulating H⁺ fluxes is demonstrated in this MS, it would be nice to show how these H⁺ fluxes alter the membrane potential in tip and shank. Based on the H⁺ flux data the tip should show depolarisation and the sank should show hyperpolarisation. The AHA mutants’ also should have depolarised membrane potential compare to wild type in shank.

Reply:

The reviewer raises an extremely important point. Indeed, the existence of such gradients of membrane potentials, while intuitively based on analyses of H⁺ flux patterns and ion

concentrations, is extremely hard to prove or even explain. Given the nature and concentration of ions, their diffusion rates, the existence of buffers for practically every ion, and the presence of fast rectifying channels throughout the cell, namely of K^+ , the theoretical consensus is that such gradients must exist, but should not pass beyond a few tens of nanometers along the plane of the plasma membrane in living cells (Savtchenko *et al.*, Nat Rev Neurol 18:598, 2017). Such standing gradients in membranes were never reliably observed within individual cells to the best of our knowledge. As of now, we have no evidence, robust to the level of publication, that pollen tubes are exceptions. As seen in the submitted new results, the sub-apical potential, as reported by ANNINE-6, seems to be slightly lower than that in the shank in the *aha6/8/9* mutant, but the difference is not statistically significant. We are actively investigating the existence of such gradients, but the absence of a physical, membrane-like division between the tip and the rest of the pollen tube remains a phenomenal conceptual barrier to the existence of such gradients. Given the strict focus on AHA physiology in this manuscript we now submit, for which we propose novel mechanistic hypotheses, we opted not to speculate on the existence of tip gradients of plasma membrane potential.

Comment by reviewer:

Some other minor points to consider

P 5 In 3, 6 and 7 the figure you are mentioning here is 1a NOT 1b.

We thank the reviewer for spotting this mistake and have corrected it in the revised version of our manuscript.

P 7 In 15-16 Indeed, overexpression of AHA7 partially rescued the pollen tube growth defect of *aha6/8/9*.- No data has been presented to support this claim.

We have deleted the above sentence from the revised version of our manuscript, and changed the preceding sentence to improve clarity: "*aha6/7/8/9* quadruple mutants were not identifiable (Supplementary Fig. 7), which suggests that AHA7 functions redundantly with the AHA proteins and weakly sustains pollen tube growth in *aha6/8/9*."

P21 In 30-34. It is critical to provide ion-selective cocktail catalogue number used for making H^+ and Cl^- selective electrodes.

We apologize for the omission; this information has now been added. For reference, Cl^- is as in Zonia *et al.*, 2002, Plant Cell, and Domingos *et al.*, 2019, New Phytol., and H^+ is as in Feijó *et al.* 1999, J.Cell Biol. or Damineli *et al.*, 2017, JXB.

P35 Figure 1 legend. Please modify based on the actual figure. Panel a and b are swapped.

We thank the reviewer for spotting this. We have corrected this error in the revised version of our manuscript.

P 36 Figure 3 legend. On panel C please mention negative sign means influx and the positive sign means efflux.

Legend was corrected.

Please revise supplementary figure 9 legend panel information.

Panel information was revised.

REVIEWERS' COMMENTS:

Reviewer #1 (Remarks to the Author):

The authors have greatly improved the manuscript and it is now acceptable for publication.

Reviewer #3 (Remarks to the Author):

Congratulations to authors for addressing all the comments raised. Really an excellent paper, I hope this will form basis to explore other roles of H⁺-ATPase.

Response to reviewer's comments

REVIEWERS' COMMENTS:

Reviewer #1 (Remarks to the Author):

The authors have greatly improved the manuscript and it is now acceptable for publication.

Response:

Thank you.

Reviewer #3 (Remarks to the Author):

Congratulations to authors for addressing all the comments raised. Really an excellent paper, I hope this will form basis to explore other roles of H⁺-ATPase.

Response:

Thank you.